# Parity and post-reproductive mortality among U.S. Black and White women: Evidence from the health and retirement study

Cheryl Elman[1]*, Angela M. O'Rand[2], Andrew S. London[3]

1 Duke University Population Research Institute and Center for Population Health and Aging, Duke University, Durham, NC, United States of America, 2 Department of Sociology and Duke University Population Research Institute and Center for Population Health and Aging, Duke University, Durham, NC, United States of America, 3 Department of Sociology, Aging Studies Institute and Center for Aging and Policy Studies, Maxwell School of Citizenship & Public Affairs, Syracuse University, Syracuse, NY, United States of America

☯ These authors contributed equally to this work.
* cheryl.elman@duke.edu

**Data Availability Statement:** Study data cannot be shared publicly because they include Restricted Data pertaining to study participants' dates and places of birth. Data are available from Health and

## Abstract

Population health research finds women's mortality risk associated with childlessness, low parity (one child), and high parity (6+ children) in a U-shaped pattern, although U.S. studies are inconsistent overall and by race/ethnicity. Parity, however, is contingent on women's biophysiological likelihood of (in)fecundity as well as voluntary control practices that limit fertility. No studies have empirically examined infecundity differentials among women and their potential contribution to the parity–post-reproductive mortality relationship or the race/ethnic-related mortality gap. We examine 7,322 non-Hispanic Black and White women, born 1920–1941, in the Health and Retirement Study, using zero-inflation methods to estimate infecundity risk and parity by race/ethnicity. We estimate proportional hazards models [$t_0$ 1992/1998, $t_1$ 2018] to examine associations of infecundity risk, parity, early-life-course health and social statuses, and post-reproductive statuses with all-cause mortality. We find Black women's infecundity probability to be twice that of White women and their expected parity 40% higher. Infecundity risk increases mortality risk for all women, but parity–post-reproductive mortality associations differ by race/ethnicity. White women with one and 5+ children (U-shaped curve) have increased mortality risk, adjusting for infecundity risk and early-life factors; further adjustment for post-reproductive health and social status attenuates all parity-related mortality risk. Black women's parity–post-reproductive mortality associations are not statistically significant. Black women's post-reproductive mortality risk is anchored in earlier-life conditions that elevate infecundity risk. Results suggest a need to focus upstream to better elucidate race/ethnic-related social determinants of reproductive health, infecundity, parity, and mortality.

Retirement Study Administrators with a Restricted Data Application and approval by the Health and Retirement Study Executive Committee for researchers who meet the criteria for access to confidential data. Contact Information: HRS Restricted Data Application Processing, Survey Research Center, P.O. Box 1248, Ann Arbor, Michigan 48106-1248. To apply for data access use: https://hrsdata.isr.umich.edu/rda/rda-application-vdi or https://hrs.isr.umich.edu/data-products/restricted-data/".

**Funding:** The first two authors received research project support from an NICHD Population Dynamics Research Infrastructure Program award to the Duke Population Research Center (P2C HD065563) and an NIA Centers on the Demography and Economics of Aging Program award to the Duke Center for Population Health and Aging (P30 AG034424) at the Duke Population Research Institute. The third author received research project support from an NIA Centers of the Demography and Economics of Aging award to the Center for Aging and Policy Studies in the Aging Studies Institute at Syracuse University (P30 AG066583). The content is solely the responsibility of the authors and does not necessarily represent the official views of the NICHD or the NIA. The funders of this study had no role in study design, data collection and analysis, decision to publish, or preparation of the manuscript.

**Competing interests:** The authors have declared that no competing interests exist.

## Introduction

Women's reproductive careers—including the number of children they bear (parity) and the life-course timing of their childbearing—contribute to post-reproductive mortality risk among midlife and older survivors. While early studies reported different strengths and directions of parity—post-reproductive mortality associations, the most recent European Census- and population registry-based studies report associations manifesting as J- or U-shaped curves [1–6]. These patterns signify that the reproductive statuses of childlessness, low parity (one child), and high parity (generally 5+ or 6+ children) are associated with elevated post-reproductive mortality risk. Studies of U.S. women, however, have not resolved inconsistent findings of: weak or no parity-mortality associations [5,7]; elevated mortality risk with higher parity [8]; a protective effect of higher parity on mortality [9]; and variable directions of associations across groups defined by race/ethnicity [10,11]. Additionally, most European studies and U.S. studies that pool Black and White subsamples report an inverse relationship between age at first birth and mortality [5,7,12–14]. However, U.S. studies that sample or stratify by race/ethnicity find that a younger age at first birth can be health-protective for Black women [10,11,15].

Inconsistent U.S. parity—post-reproductive mortality findings are not well-addressed in the literature and may reflect selectivity in childlessness status due to infecundity and other selective processes. In this paper, we refer to (in)fecundity as the (in)ability to reproduce, and refer to fertility or observed parity as the number of (live) children born [16]. Infecundity refers to the physiological incapacity to bear live children. Childless women (zero observed births) pose a problem in parity—post-reproductive mortality studies because they are a heterogeneous group: they might have been sorted into nulliparity due to their own biologically-based infecundity or other factors, such as voluntary control of childbearing, nonmarital celibacy, or spousal infertility [16–20]. This problem may underlie inconsistent U.S. findings: it is under-appreciated that pre-1940s birth cohorts of U.S. Black women had both higher fertility rates and higher childlessness rates than White age peers, with the latter partially attributable to poorer reproductive health and infecundity [20–23]. To better understand the parity—post-reproductive mortality relationship, we examine fertility as a process, where observed parity is contingent on women's differential probabilities of (in)fecundity (their biophysiological likelihood of bearing children) associated with early-life-course factors, including health and social environments [17,18,24]. Social environmental factors are also associated with race/ethnicity-related health inequities [25]. Consequently, selectivity in childlessness due to infecundity may differentially shape parity—post-reproductive mortality associations and contribute to race/ethnic-related differences in associations net of other social determinants of health, parity, and mortality.

Using data from the 1992–1998 waves of the Health and Retirement Study (HRS), we follow 1920–1941 birth cohorts of non-Hispanic Black and White women through 2018. We first use zero-inflation count models to examine women's probability of membership in a latent infecund class. We then use proportional hazards models to examine the associations between women's probability of infecundity, observed parity categories, and all-cause mortality. In analyses focused on parous women, we examine the associations of timing variables—age at first birth and premarital birth—with all-cause mortality. In all analyses, we examine race/ethnic-related differences in these associations, accounting for infecundity risk. Because social and health selection processes occurring over the life course contribute to the parity—post-reproductive mortality association [7,19,26], we sequentially adjust for place-of-birth factors, early-life-course health and socioeconomic statuses, and adult (post-reproductive) socioeconomic, marital, and health statuses in the nested proportional hazards models we estimate.

## Factors contributing to parity–mortality associations

### The childbearing context

The characteristics of populations, including their historical settings, contribute to observed parity—post-reproductive mortality associations [23]. For example, an early meta-analysis of 31 studies of women by Hurt and colleagues found one strong relationship among otherwise inconsistent ones [26]. In twelve of thirteen historical cohorts studied, mortality was highest among childless and low-parity women and declined with parity; contemporary birth cohorts, in contrast, more often exhibited U- or J-shaped parity-mortality curves. Subsequent meta-analyses also report U- or J-shaped curves in recent but not older birth cohorts [27,28]. Different associations by birth cohort partly reflect changes in childbearing contexts. The historical birth cohorts in Hurt et al.'s study [26] were high-fertility populations exposed to high rates of infant mortality. In this context, childless women were more likely childless involuntarily and not because of conscious, voluntary control of childbearing that could threaten community (and personal old-age) survival, religious beliefs, and/or other societal norms. Populations that do and do not practice fertility control are distinctive and observed parity—post-reproductive mortality associations likely differ on this basis [19,26].

This distinction has implications for U.S. parity-post–reproductive mortality research. Most U.S. studies sample early- to mid-twentieth century U.S. birth cohorts. Women in these historical birth cohorts shared atypical marriage and childbearing patterns; many could have given birth between 1946–1964, thereby contributing to the historical Baby Boom. Unlike women in contemporary birth cohorts, they more likely married, married prior to age 25, preferred to bear at least one child, had higher completed fertility, and primarily gave birth within marriage [7,29]. Indeed, one advantage of studying historical populations marked by a high prevalence of early marriage and marital childbearing, is that the parity—post-reproductive mortality relationship is less likely to be masked. Another advantage is that the potential selectivity surrounding marital childbearing that occurs in contemporary birth cohorts is minimized [7,12].

Yet, U.S. studies have overlooked U.S. childbearing contexts—and associated reproductive behaviors—that diverged by Census region and race/ethnicity. For example, the marital fertility rates of Black and White women in the American South, through the early 1900s, resembled levels found in populations that do not voluntarily control their fertility (i.e., natural fertility populations) [21]. Comparatively higher southern fertility rates then persisted to the 1940s [30]. Fertility control practices, however, diverged more by race/ethnicity than region [21,22]. Southern White women by 1900, like White women in all U.S. regions, show evidence of fertility control by both limiting marriage (nuptiality) and marital fertility [21,29]. Southern Black women compared to all White women, to the 1940s, more likely married, remarried, had higher fertility rates, and less-likely limited marital childbearing [20,21,22,31]. Their reproductive practices reflected their distinctive childbearing contexts: about 95% of U.S. Black women in 1900 (dropping to 75% in 1940) lived in the South and larger family sizes benefited their predominantly agricultural, but not land-owning, household economies [32,33].

Yet, Black women born between 1880 and 1940 had remarkably higher rates of childlessness than White age peers. A crossover in childlessness rates first occurred in mid-1880s U.S. birth cohorts: about 22% of White and Black women had remained childless by midlife, although Black women's rates were rising and White women's rates were declining [34–37]. About 20% of White women in the 1909 birth cohort remained childless, falling to about 6–7% in 1924–1929 birth cohorts [36,37]. In contrast, about 30% of Black women in 1909 to 1924 birth cohorts remained childless, falling to about 15%—twice the level of White women—in early 1930s birth cohorts [35,36,37]. A second crossover emerged in the 1942 birth cohort

(approximately) as White women's childlessness rates rose to surpass Black women's rates, producing a reversal in the U.S. race/ethnic-related gap in childlessness [35,36,37]. It is important, however, that forces underlying childlessness rates differed by historical period and race/ethnicity. The initial 1880s crossover and new race/ethnic-related gap reflected Black women's poorer reproductive health [17,22,33,34]. The 1940s crossover and reversed race/ethnic-related gap reflected Black women's falling childlessness rates due to improving health [17,22] and White women's rising rates, due to greater adoption of conscious fertility control and two-child fertility and childlessness norms [29,37,38].

Studies that examine parity progression further suggest divergent race/ethnic-related fertility control patterns in 1880–1940 U.S. birth cohorts. They reveal that the higher fertility-higher childlessness pattern in 1880–1940 birth cohorts of Black women manifested as a fecundity threshold, such that parous Black women able to have at least one child more likely had a next birth with each succeeding birth (i.e., no evidence of stopping at a particular parity) [34,35]. In contrast, parous White women in the same birth cohorts were less likely to have a next birth with each succeeding birth (i.e., evidence of stopping or voluntary limitation) [34,35].

These period, Census region, and race/ethnic-related differences in fertility patterns provide evidence that biophysiological in conjunction with social-environmental factors anchor parity—post-reproductive mortality associations [1]. The latter factors shaping childbearing contexts include: economic development and related factors that improve survival environments [6,39]; shifting policies and norms about women's education, work, and family formation [27,40]; structural and regional race/ethnic-related inequalities [25]; and societal norms about childbearing limitation [38]. We address these factors further, below.

## Biophysiological factors

Disposable soma and other evolutionary frameworks motivate much parity-mortality research. Researchers posit that the greater biological impetus to reproduce, rather than to maintain physiological fitness, comes with a biophysiological cost or trade-off: higher parity should shorten, and lower parity lengthen lifespans [41]. Although evolutionary pressures are difficult to empirically isolate in human studies, there is evidence of this expected trade-off in studies using multi-generational data within homogeneous (elite) populations [41]. Some contemporary Census-based studies also report this expected trade-off [12], while others do not [2,42]. However, meta-analytic studies find childlessness and lower parity associated with shorter, not longer, lives—the opposite of the expected trade-off—in historical high-fertility, high-mortality populations [26]. Recent studies examining this trade-off in the context of modern, higher living standards, reduced infectious disease, and reduced risk of early-life mortality—better survival environments—also yield inconsistent results [6,39]. In such contexts, however, more highly-resourced populations should be better able to overcome biophysiological constraints [6].

Importantly, interpretation of all findings from these studies is complicated by the inconsistent inclusion of nulliparous women across analytic samples; their exclusion can obscure left-hand portions of J- or U-shaped distributions. Studies also elide differences in childbearing contexts, especially their influence on the prevalence of voluntary limitation practices [19,26]. No study empirically distinguishes involuntary and voluntary childlessness and therefore, by default, all studies combine the infecund and the voluntarily childless, if they include the childless at all. Together, these factors may account for a lack of an expected trade-off pattern. As such, a default selective "healthy pregnant woman effect" may underlie observed historical population patterns: only reproductive-aged women with the physiological resilience to have reproductive ability *and* survive exposure to infectious diseases, malnutrition, and other health risks, including actual childbearing, reach higher parities and post-reproductive age [12,19,26].

## Social environmental factors

Maternal depletion frameworks highlight the reproductive versus biophysiological/ metabolic trade-off but conceptualize maternal childbearing as resource-depleting or resource-neutral, not resource-enhancing. Maternal resource repletion between births or after the completion of childbearing is possible, albeit variably contingent on community, social, and familial resources [43,44]. Generally, it occurs more in resource-rich and healthful environments than in resource-poor and/or high-reproductive-risk environments [43,44,45].

Additionally, and of special relevance to our study, repletion is least likely when women's reproductive careers are embedded in lifetime trajectories of structural disadvantage that produce chronic physiologically stressful conditions [45]. Critically, higher childlessness rates in 1880–1940 birth cohorts of U.S. Black women reflected childbearing contexts marked by greater exposure to infectious and nutritional diseases and environmental health risks, such as poor housing quality, compared to White age peers [22,23,34]. Southern Black childbearing-aged women had twice or higher mortality rates from southern infectious and nutritional diseases, such as malaria and pellagra, and from tuberculosis and venereal diseases [17,46–48]. Among survivors, the cumulative effect of these exposures across the life course, amidst structural racism as practiced in the Jim Crow South [49,50], would significantly impair health, including processes of maternal repletion.

Social determinants of health [51] and "weathering" frameworks focus attention on the embodiment of social disadvantage. Weathering conceptualizes the emergence of global health deficits (i.e., not limited to reproduction) as a lifelong process of: accumulated and accelerated childhood physical maturation, embodied as the early onset of menarche [52]; high rates of pregnancy complications associated with poorer maternal and fetal health [15]; premature aging associated with high chronic disease prevalence rates at midlife [53]; and reduced longevity [11,54]. Weathering implicates social and biophysiological mechanisms, including disadvantaged family origins, individual experiences of adversity, and low socioeconomic status in adulthood [11]. In most U.S. reproductive health studies, even with these factors controlled, race/ethnicity—and by this we mean unmeasured factors associated with race/ethnicity [25]—remains significant [10,11,14].

Studies testing social integration perspectives find parity-health and -mortality relationships influenced by and through social networks of support, including families of origin, spouses or partners, adult children, and fictive kin [3,7,9,27,55]. The identified mechanisms by which kin support is beneficial include financial and instrumental help and/or emotional support [43,55]. For example, greater longevity among couples with higher fertility may reflect influences of long-term companionship, social integration, and support from adult children [9,55]. Greater risk of post-reproductive mortality among the childless may reflect social norms that privilege women as mothers over the life course, inclusive of social and/or economic supports at the end of life [40]. Alternatively, it may be that the health behaviors of those with higher-fertility become more health-protective with increased childrearing experience [9]. Statistical controls for social factors, among the parous, can substantially attenuate or reduce parity—post-reproductive mortality associations to non-significance [40,55]. However kin support—and the need for support—likely varied by whether childlessness resulted from involuntary versus voluntary (i.e., planful) circumstances.

Influences of socioeconomic and health factors on parity—post-reproductive mortality associations reflect social selection as well as social causation. Families of origin differ in their capacity to provide education, foster health, provide nutritious foods, and socialize children about life roles, including parenting and the expected timing and sequencing of marriage and childbearing [38,40]. Some of the factors noted above (e.g., childhood poverty, poor living

standards) are likely associated with both childlessness and high fertility, albeit though different mechanisms. They also are likely to vary in relation to race/ethnicity. While early parity-mortality studies did not always theorize or adequately measure social and health selection [7,26], current studies that adjust for these factors find that selective processes cumulatively exert their influence from childhood through adulthood and account for a portion of the parity—post-reproductive-mortality association [1,5,7,11]. These studies advance our understanding of the links between parity and mortality, but no study has examined whether health and social selectivity that is sufficiently severe to preclude childbearing is a contributing factor.

### Research aims

The mortality risk of childless women may exceed that of all other women [26,27,28]. However, studies often sidestep this issue, as a result of study design, by excluding nulliparous women from their analyses. No study to date has examined the possibility of heterogeneous associations between involuntary and voluntary childlessness and post-reproductive mortality, or whether a consideration of involuntary childlessness can contribute to our understanding of race/ethnic-related differences in parity—post-reproductive mortality associations.

The current study has three aims that address these issues. First, we examine women's probability of infecundity (i.e., involuntary childlessness) in a sample of non-Hispanic Black and White women born between 1920 and 1941. A first hypothesis is that:

*Hypothesis 1*: We expect to find that Black women have a higher probability of infecundity, but, at the same time, an equal or higher mean number of births relative to White women.

Second, we examine whether women's probability of infecundity plays a substantively important role in parity—post-reproductive-mortality associations, net of the childhood and adulthood socioeconomic and health contexts that prevailed for Black and White women in these historical birth cohorts. Specifically:

*Hypothesis 2*: We expect to find a U-shaped parity-mortality association such that infecundity risk, low parity, and high parity, relative to 2 births, are positively associated with mortality.

*Hypothesis 3*: We expect to find that the life-course timing of childbearing is associated with mortality such that premarital birth, and younger and older age at first birth, elevate mortality risk.

Finally, U.S. scholars have long argued that poorer health and living standards contributed to higher Black than White rates of early twentieth-century childlessness [17,23,34,35]. We examine whether race/ethnic-related differences in parity—post-reproductive mortality associations remain after adjusting for the probability of infecundity, and selected variables measuring women's childhood and adulthood health and socioeconomic statuses.

*Hypothesis 4*: We expect to find that race-related differences in mortality risk are only partially explained by parity and its timing, such that Black women will have greater post-reproductive mortality risk than White women.

## Material and methods

### Data and sample

We use the Health and Retirement Study (HRS), a nationally representative longitudinal study of U.S. adults. The first wave, conducted in 1992, interviewed persons born between 1931 and 1941 (ages 51–61 years); second and third waves were fielded in 1994 and 1996, respectively. A

companion study conducted in 1993, the Study of Asset and Health Dynamics of the Oldest Old (AHEAD), interviewed persons born in 1924 or before; a second wave was fielded in 1995. In 1998, HRS and AHEAD cohorts were merged, and the Children of the Great Depression (CODA) cohort, born between 1925 and 1930, was added. HRS and AHEAD cohorts over-sampled Black adults and the 1992 screener used to generate the initial HRS and AHEAD cohorts oversampled Florida residents. HRS and AHEAD response rates were 81.1% in 1992/ 1993, 90.7% in 1994/1995, 86.9% in 1996, and 83.8% in 1998 [56].

This study's main HRS data source is the RAND Longitudinal 1992–2018 (Version 2) database (hereafter RAND). The RAND provides harmonized responses for selected measures across all HRS and AHEAD waves. We linked the RAND to two HRS databases, the Cross-Wave Geographic Information (State) Restricted Data File (1992–2020) and the Exit Date of Death Restricted File (1992–2020), to obtain participants' states of birth and dates of death, respectively. We additionally linked the RAND to the: RAND Family Respondent File (1992–2014); RAND Detailed Imputation File (1992–2020); HRS 2020 Tracker File; and CORE-Demography files in each of the 1992, 1994, 1996, 1998 HRS and 1993, 1995 AHEAD waves. We limited analyses to women who self-identified as non-Hispanic Black or White, were born between 1920 and 1941, and entered the sample between 1992–1998. The lower birth cohort bound (1920) avoids population health anomalies associated with the U.S. influenza epidemic (1918–1919) and the upper bound (1941) avoids the 1940s race/ethnic-related shift in childlessness [35,36].

We linked the RAND data to Eriksson et al.'s [57] Revised Infant Mortality Rates and Births for the United States, 1915–1940. Eriksson et al.'s (2018) database provides better historical estimates of live births (denominators) than available in vital statistics data, although it uses unaltered infant deaths (numerators) taken from vital statistics data. We use Adjustment 4 that accounts for migration and other sources of lumpiness in birth rate data across states/ time. We linked the IMR for each participant's year and state of birth to her record using the appropriate birthplace Federal Information Processing Standards (FIPS) code provided in the HRS Restricted Geographic Information File. We excluded from the analytic sample women lacking a state FIPS code and women with missing data on early-life-course marital statuses and baseline health conditions (see S1 Table). The final analytic sample includes 7,322 women (unweighted). The Duke University Institutional Research Board approved this study's proto-col (#2019–0641).

### Dependent variables

**Fertility.** We obtained the first dependent variable, number of *Children Ever Born*, from the RAND. The HRS asked participants about their number of live births at intake interviews (HRS 1992–1998; AHEAD 1993–1995). About 6% of the sample had missing values for the RAND harmonized variable; we used a second variable, participant's number of *Own (living) Children*, taken from the RAND Family Respondent File, to fill in missing values and flagged this substitution in the count regression models we estimated. The final distribution of the *Children Ever Born* variable had ten outliers; to address this, we assigned women with 10 or more children a value of 10.

**All-cause mortality.** We followed participants to December 31, 2018, using the HRS Restricted Mortality File (RM), 2020 Tracker File, and RAND in-wave status variables to determine survival, death, and attrition statuses. Fifty-nine percent of participants died over the follow-up period; we consider those alive on the follow-up date to be right-censored. About 9% of participants left the sample prior to this date. From Tracker-File data, we assigned each attritor a date of death or date when last known to be alive and treated those observations as right-censored (see Analytic Strategy below).

## Primary independent measures

**Observed parity and timing indicators.** Using RAND Family Respondent File (1992–2014) data to determine *Age at First Birth*, we subtracted the age of each participant's self-reported oldest own child at the intake interview from the participant's age at the intake interview if both were reported. (This database provides information only about each participant's oldest and youngest currently-living child with whom they were in contact at the intake interview, not all children born.) If an oldest own child's age was missing at the intake wave, we used the oldest own child age reported in the next available wave, subtracting it from the participant's age at that wave. Consistent with Henretta [7], we considered women 20 years of age or younger at first birth to have an *Early Age at First Birth* and women with first births after age 35 years to have a *Late Age at First Birth*. We considered women whose age at first birth was less than their marital age to have had a *Premarital Birth*. Notably, HRS/AHEAD women were not queried about cohabiting unions; a cohabiting union pre-dating a reported first marriage may have produced reported children. Also, RAND Family data on children's ages are not uniformly reliable [7].

## Adjustment measures

**Demographic characteristics.** Participant *Age* is year of age at the intake interview, taken from the HRS 2020 Tracker File. *Race/Ethnicity*, self-reported, is non-Hispanic Black, with non-Hispanic White as the reference.

**Birthplace characteristics.** We examined parity-mortality associations in the context of birthplace health environment, as indicated by the *Infant Mortality Rate* (IMR) in each participant's state and year of birth. The IMR is a robust measure of population health, aggregate income level, and living standards [11]. When participant IMR was missing due to non-reporting birth state, we imputed state/year values from 20 imputed datasets [58]. We used a dichotomous variable to indicate participants *Born in the South* (as defined by the U.S. Census). Birth in another U.S. region is the reference.

**Childhood socioeconomic and health status.** RAND harmonizes mother and father educational status indicators across waves. Parent education for AHEAD cohorts referenced parent(s) having an 8th grade attainment level or not. To maintain HRS/AHEAD consistency, we distinguished, for all participants, *Parent Education* categorically: father attaining 8th grade or higher, missing, with attaining less than 8th grade as the reference. We filled in missing values with mother's attainment relative to completing 8th grade.

Participants in the 1998 HRS wave reported health status to age 16, ranging from excellent (1) to poor (5). We constructed categorical *Self-Rated Childhood Health*, with categories of poor/fair and missing; good/very good/ excellent health is the reference.

*Number of Living Siblings*, harmonized in the RAND from intake interviews, is an ordinal indicator of each participant's mother's reproductive fitness (i.e., biophysiological factors) and participant's childhood socialization about normative family size (i.e., social environmental factors).

**Marital histories.** *Marital Duration*, in years, is obtained from a RAND harmonized variable reporting the duration of a woman's longest marriage. *Age at Marriage* denotes, in this era, the normative initiation of exposure to the risk of pregnancy. HRS and AHEAD developed different algorithms to query participants about marital histories. At the baseline interview, about 72% of women in our sample reported a single marriage, 26% reported a higher-order marriage, and 2.9% reported being never-married. To construct *Age at Marriage* for the first-married, we subtracted each participant's year of birth from their year of marriage, if reported. If year of marriage was missing, we subtracted baseline *Longest Marital Duration* (given in

years in the RAND) from the interview year to derive year of marriage; we then subtracted year of birth from this value. For those in higher-order marriages, we used the earliest reported year of marriage in women's intake records and subtracted year of birth from this value. For AHEAD widowed/ divorced/cohabiting participants, we used the reported year that their marriage ended and its duration in years to derive *Age at Marriage*. Preliminary analysis of the joint distributions of *Age at Marriage* and *Children Ever Born* indicated that almost a third of the 212 never-married women had children and nearly 20% had two or more children. HRS/ AHEAD women were not queried about non-marital unions, known to be associated with model variables of race/ethnicity, social class, health resources, and birth timing. Rather than exclude them, we assigned never-married women the mean *Age at Marriage* for the sample, flagging the substitution in the *Children Ever Born* count analyses. Sensitivity analyses that (1) used age at marriage categories including never-married and (2) excluded all never-married women produced the same substantive results as those we report in the paper.

**Adult socioeconomic status.** Most U.S. women born before 1941 completed education prior to the onset of childbearing. However, the 1960s U.S. Manpower Act and economic growth fueled a boom in adult education, including for middle-aged women whose children had aged out of the home [59]. We consider completed *Education* at the intake interview as adult socioeconomic status; we used a RAND harmonized education measure to form categories of less than high school and greater than high school, with high school graduate/GED serving as the reference category. We additionally created measures of baseline *Household Income* (logged) and *Home Ownership*, a dichotomy, using RAND Detailed Imputation File data. Preliminary analyses of mortality included baseline household wealth, but inclusion of this measure did not improve model fit. Thus, we do not adjust for household wealth.

**Adult health status and health behaviors.** A variable, *Number of Medical Conditions*, taken from the RAND, indicates whether participants at intake interviews ever had physician-diagnosed high blood pressure, diabetes, heart disease, stroke, lung disease, cancer, or arthritis. We also adjust for health behaviors [60]. *Ever Smoked*, observed at the intake wave, is a dichotomous variable indicating whether a woman ever smoked. *Alcohol Use* was ascertained by different questions across intake waves. We coded heavy drinking as having 3+ drinks/day for HRS 1992 and AHEAD 1993 participants and as drinking 5 or more days a week for HRS 1994–1998 and AHEAD 1995 participants.

## Analytic strategy

Table 1 reports sample characteristics as proportions for categorical variables and means and standard deviations for continuous variables. All statistical analyses in Tables 1–5 report significance tests based on 95% confidence intervals.

### Fertility count models and infecundity risk

The first dependent variable in our study, Number of Children Born, is a count. Count regression methods are recommended for modeling count dependent variables as they have non-normal distributions. A first-choice model, a Poisson count model, assumes relative homogeneity in a study population (assumes that count equation variance is equal to its mean), although study populations' characteristics might vary. Statistical tests of Poisson model fit to the data allow this assumption to be rejected. If so, (e.g., if there is a great deal of error variance or dispersion in the Poisson model), a next step is to estimate a negative binomial count model that relaxes the Poisson variance-equal-to-the-mean assumption.

Preliminary analysis of our sample showed that 10.5% of Black women and 8.6% of White women had zero births. We knew that, in theory, childlessness (zero births) among sample

**Table 1. Descriptive information by race/ethnicity.**

| | Total (N = 7,322) | Black (N = 1,274) | White (N = 6,048) | P-Value |
|---|---|---|---|---|
| *Outcome-Related Variables* | | | | |
| Risk Period (Days) (Mean/S.D.) | 6792.26 (3133.42) | 6607.41 (3278.80) | 6831.20 (3100.81) | .02 |
| Mortality (N/%) | 4358 (59.5) | 805 (63.2) | 3553 (58.8) | .003 |
| Attrited (N/%) | 694 (9.5) | 90 (7.1) | 604 (10.0) | .001 |
| *Sociodemographic N/%* | | | | |
| Black | 1274 (17.4) | | | |
| Age (Mean/SD) | 61.39 (7.66) | 59.85 (7.20) | 61.71 (7.71) | < .001 |
| Age 50–59 | 3701 (50.6) | 757 (59.4) | 2,944 (48.7) | < .001 |
| Age 60–69 | 1750 (23.9) | 273 (21.4) | 1477 (24.4) | < .001 |
| Age > = 70 | 1871 (25.6) | 244 (19.2) | 1627 (26.9) | < .001 |
| *Parity N/%* | | | | |
| Number of Children Born | 3.07 (2.01) | 3.79 (2.73) | 2.92 (1.78) | < .001 |
| Zero Reported Children | 655 (8.9) | 134 (10.5) | 521 (8.6) | .03 |
| One Child | 727 (9.9) | 156 (12.2) | 571 (9.4) | .002 |
| Two Children | 1760 (24.0) | 186 (14.6) | 1,574 (26.0) | < .001 |
| Three Children | 1637 (22.4) | 181 (14.2) | 1,456 (24.1) | < .001 |
| Four Children | 1122 (15.3) | 171 (13.4) | 951 (15.7) | .038 |
| Five Children | 624 (8.5) | 124 (9.7) | 500 (8.3) | .088 |
| Six + Children | 797 (10.9) | 322 (25.3) | 475 (7.8) | < .001 |
| *Marital History N/%* | | | | |
| Age at Marriage (Mean/SD) | 23.66 (8.71) | 25.86 (10.49) | 23.22 (8.24) | < .001 |
| Marital Duration | 35.43 (13.20) | 30.40 (13.70) | 36.48 (12.84) | < .001 |
| #Mother Births (#Sibs) (M/SD) | 2.65 (2.25) | 3.54 (2.80) | 2.46 (2.06) | < .001 |
| Ever Remarried | 1869 (25.5) | 319 (25.0) | 1550 (25.6) | .66 |
| Early First Birth | 3786 (51.7) | 823 (64.6) | 2963 (49.0) | < .001 |
| Late First Birth | 106 (1.4) | 15 (1.2) | 91 (1.5) | .25 |
| Premarital Birth | 2948 (40.3) | 678 (53.2) | 2270 (37.5) | < .001 |
| *Birthplace, Childhood N/%* | | | | |
| Infant Mortality Rate (Mean/SD) | 56.74 (14.61) | 59.63 (12.43) | 56.14 (14.96) | < .001 |
| Born South | 2978 (40.7) | 1084 (85.1) | 1894 (31.3) | < .001 |
| Parent Less than 8th Grade | 1772 (24.2) | 473 (37.1) | 1299 (21.5) | < .001 |
| Parent: 8th Grade or Higher | 4977 (68.0) | 628 (49.3) | 4349 (71.9) | < .001 |
| Child Health Poor/Fair | 423 (5.8) | 87 (6.8) | 336 (5.6) | .07 |
| Child Health Good to Excellent | 5924 (80.9) | 963 (75.6) | 4961 (82.0) | < .001 |
| *Adulthood N/%* | | | | |
| Less than High School | 1822 (24.9) | 568 (44.6) | 1254 (20.7) | <001 |
| High School | 4290 (58.6) | 546 (42.9) | 3744 (61.9) | < .001 |
| Greater than High School | 1210 (16.5) | 160 (12.6) | 1050 (17.4) | < .001 |
| Lives in the South | 2979 (40.7) | 696 (54.6) | 2283 (37.8) | < .001 |
| Owns Home | 5993 (81.8) | 824 (64.7) | 5169 (85.5) | < .001 |
| Income ($) (Mean/SD) | 40008 (73052.9) | 23966.4 (23885) | 43388.3 (79217) | < .001 |
| Baseline Married | 4955 (67.7) | 599 (47.0) | 4356 (72.0) | < .001 |
| # Health Conditions (Mean/SD) | 0.64 (0.98) | 1.0 (1.16) | 0.57 (0.92) | < .001 |
| Heavy Drinking | 214 (2.9) | 12 (0.9) | 202 (3.3) | < .001 |
| Ever Smoked | 3838 (52.4) | 685 (53.8) | 3153 (52.1) | .29 |

Note: Chi-square tests were used to determine whether categorical variables differed by race/ethnicity.

t-tests were used to determine whether continuous variables differed by race/ethnicity.

women might have reflected prior biophysiological (infecundity) or voluntary limitation processes. Moreover, if two different underlying processes (infecundity versus childbearing control) had produced zero births, two different groups of childless women would be present in our sample. The childless women from different groups could "inflate" the zeros in the fertility count. A negative binomial count model, used to relax the assumption of population homogeneity, might still not provide an adequate or best fit to the data.

That different underlying processes can produce different groups of childless women poses the problem of distinguishing the infecund from voluntarily childless nulliparous women. Fertility studies address this problem by additionally testing fit to the data of zero-inflation Poisson (ZIP) and zero-inflation negative binomial (ZINB) count models [61,62]. These models estimate the sample women's probabilistic membership in one of two possible latent groups: an "always-zero" or infecund group and a group where women might have zero births but do not statistically fall into the always-zero profile [61–64]. Specifically, zero inflation models estimate a count of women's observed number of children born as a function of two distinct processes: (1) a logistic (or probit) process that distinguishes a possible latent, always-zero group of probabilistically infecund women; and (2) a count process that estimates the parity distribution among probably fecund women, also allowing for zero births [62,63]. Zero-inflation models allow evaluation of the hypothesis that there is a latent always-zero (i.e., infecund) class of women in the data.

Using Proc Genmod, in SAS 9.4, we modeled women's *Number of Children Born* as a function of: age; age at marriage (polynomial), marital duration [64], and number of siblings and race/ethnicity. We initially used Poisson count models. Finding considerable dispersion, we additionally examined negative binomial and ZIP and ZINB models. We compared, across regression models, model fit to the data using: ratios of Pearson chi-square and deviance scores to degrees of freedom to test overdispersion; and Clarke tests to determine the presence/absence of a distinct latent always-zero class of probabilistically infecund women. We also examined overall model fit and parsimony of fit by using plots of predicted count distributions against the observed count and BIC scores, respectively. To preview findings, the ZINB functional form best fits the data and identify women with a statistically high probability of selection into an always-zero parity class. In the sample, the correlation between the probability of selection into an always-zero parity latent class and an observed zero parity (childless) category is .27; the $R^2$ (% of variance explained) is 7.3% (0.27 x 0.27 = 0.073). After standardizing the distribution (mean = 0, SD = 1), we exported each woman's probability of being in an always-zero parity class as a variable, *Infecundity*, and included *Infecundity* in the proportional hazard models to examine its association, alongside the association of observed parity, with all-cause mortality.

## Mortality: Proportional hazards models

In estimating all-cause mortality, we modeled sample attrition as a cause-specific competing risk, treating it as right-censoring. We first conducted preliminary tests of the proportional hazards assumption for all model variables in the full sample and in the race/ethnic-specific subsamples by examining: (1) correlations between Schoenfeld residuals for each variable and the ranked order of failure time among those who died; and (2) interactions of each variable with time. Both tests indicated non-violation of the proportional hazards assumption, except for race/ethnicity (pooled sample) and both linear and categorized age (pooled and race/ethnic-specific samples). Further examination of full-sample mortality equations also revealed statistically significant interactions of race*linear age and race*categorical age.

Due to evidence of nonproportionality by race/ethnicity, linear age, and age group, we adopted age-stratified proportional hazards models to examine mortality in the full sample of

women and in race/ethnic-specific subsamples. A graphical analysis of age group log-log curves generated from the equations in pooled and race/ethnic-specific samples indicated that the age-group strata were approximately proportional [65]. S2 Table reports full-sample results using best-fit linear age model adjustments instead of age-group stratification; parity—post-reproductive mortality findings do not substantively differ from age-stratified models.

In full-sample (Table 3; full results in S3 Table) and race/ethnic-specific analyses (Tables 4 and 5; full results in S4 and S5 Tables, respectively), we further differentiated parous plus nulliparous women (all women) from parous women. In all-women analyses, we tested parity—post-reproductive mortality associations using *Infecundity* probabilities and observed parity categories of 0, 1, 3, 4, 5, and 6+, with 2 births as the reference category for observed parity. In parous women analyses, we tested parity-mortality associations using the observed parities of 1, 3, 4, 5, 6+, with 2 births as the reference, and included early age of first birth, late age at first birth, and premarital birth. We report the results of likelihood ratio statistical significance tests for observed parity categories, as a group, in all models.

In the full sample of women (Table 3), age-stratified Model 1 examines observed parity only. Age-stratified Model 2, for parous plus nulliparous women, examines women's probability of *Infecundity* and observed parity. Age-stratified Model 5, for parous women, examines the main effects of the life-course timing and observed parity variables. It is also important to adjust parity-mortality associations for health and socioeconomic selection processes that operate over the life course and are associated with both parity and mortality [7,11]. Accordingly, age-stratified Models 3 and 6 adjust for race/ethnicity, birth place, and early life-course health and socioeconomic statuses. Models 4 and 7 further adjust for adult socioeconomic, health, health behavior, and marital statuses. We followed this same procedure in race/ethnic-specific analyses (Tables 4 and 5).

Information about children ever born, harmonized in RAND, was directly reported by HRS/AHEAD participants. In contrast, the RAND Family Respondent File ascertained participants' number of in-contact, alive children from household and participant records. As HRS/RAND data do not report children surviving in the context of children ever born, and RAND Family data do not report deaths/death dates of children who died prior to HRS/AHEAD at baseline, there is a measurement gap: given the universe of children ever born, we only have information about surviving children in late-life (RAND Family Respondent File). This gap suggests that, among parous women, age of oldest child in RAND Family data may be misstated. This could introduce error in age at birth findings, contingent on whether even earlier-born child(ren) had died/lost contact (i.e., were not present in the RAND Family roster). In a sensitivity analysis (see S6 Table), we examined possible measurement error in age at first birth variables used in Models for parous women. We created a Flag variable that differentiated number of children ever born and number of own alive, in-contact children in late-life. The Flag = 1 applied if women's number of children born *differed* from their number of in-contact, living children in late-life or = 0 if the number of children *matched*. We found that statistical results (the size and significance of hazard ratios) for age at first birth variables did not change when Flags were included in Models. All models in this study use cluster-robust sandwich standard errors due to birth-state clustering (state FIPS).

## Results

### Descriptive statistics

Table 1 describes the analytic sample. The sample is 17.4% Black and has a mean age of 61.39 years. The mean number of children born is 3.07; 9% of women were childless and almost 11% had borne 6 or more children. Parity distributions differed by race/ethnicity; more Black

women were childless (10.5% versus 8.6%), gave birth to 1 child (12.2% versus 9.4%), or had 6 + children (25.3% versus 7.8%). Additionally, Black women more likely had a first birth at age 20 or less (64.6% versus 49.0%) and a premarital birth (53.2% versus 37.5%).

## Count model

Table 2 presents ZINB estimates of *Number of Children Born*. The count model (top of the column) displays results as beta coefficients and exponentiated incidence rate ratios (IRRs). The count model finds that an older age at marriage—which would reduce exposure time to the risk of pregnancy within marriage—is associated with lower parity although marital age was less influential at older ages as indicated by the negative polynomial term. A squared term for marital duration does not improve model fit. Black women, net of other measures, have higher fertility; their expected number of children is 40% higher than White women's expected number of children (Black beta $\exp^{.34} = 1.40$). The estimated marginal mean number of children for Black women (i.e., least square mean based on all estimated effects in the count model) is 4.11; the estimated marginal mean for White women is 2.91 (not in Table). Women with higher-parity mothers (i.e., with more siblings) had more children, perhaps reflecting

**Table 2. Number of children born: Zero-inflation negative binomial model.**

|  | Count Model | | | |
|---|---|---|---|---|
|  | *beta* | *SE* | *IRR* | *95%CI* |
| Black Women | 0.34*** | 0.019 | 1.40 | [1.36–1.46] |
| *Own-Mother Fertility History* | | | | |
| Number of Siblings | 0.02*** | 0.003 | 1.02 | [1.02–1.03] |
| *Marital Exposure to Pregnancy* | | | | |
| Age at Marriage (centered) | -0.02*** | 0.002 | 0.98 | [0.98–0.98] |
| Age at Marriage Squared | 0.06*** | 0.006 | 1.06 | [1.05–1.07] |
| Marital Duration | 0.16* | 0.071 | 1.17 | [1.02–1.34] |
| Ever Remarried | -0.03 | 0.020 | 0.97 | [0.94–1.01] |
| Flag: Parity from RAND Family | -0.01 | 0.030 | 0.99 | [0.94–1.05] |
| Flag: Missing Age at Marriage | -0.90*** | 0.089 | 0.41 | [0.34–0.48] |
| Intercept | 0.99*** | 0.014 | | |
| Alpha | 0.014*** | 0.006 | | |
|  | Always-Zero Model | | | |
|  | *Logodds* | *SE* | *OR* | *95%CI* |
| Black Women | 0.67*** | 0.257 | 1.95 | [1.18–3.23] |
| Age at Baseline | 0.06*** | 0.015 | 1.07 | [1.04–1.10] |
| Age at Marriage (Centered) | 0.14*** | 0.022 | 1.15 | [1.10–1.20] |
| Age at Marriage Squared | -0.25*** | 0.061 | 0.78 | [0.69–0.88] |
| Intercept | 3.84*** | 0.181 | | |
| ZINB Correlation: | 0.091 | | | |
| Observed/Predicted 0 ($\rho\_0$) | | | | |
| BIC | 28891.8 | | | |
| Scaled Pearson Chi Sq./df | 0.992 | | | |
| N | 7322 | | | |

*$p < .05$
**$p < .01$
***$p < .001$ (two-tailed tests).

reproductive fitness and/or greater childhood exposure to religious or social norms predisposing to larger family sizes and/or avoidance of fertility control practices [40].

The zero-inflation portion of the ZINB model (Always-Zero Model) estimates women's probability of selection into a latent always-zero class. Black women's probabilities are twice that of White women's: the mean predicted probability is 0.029 for White women and 0.074 for Black women (not shown in Table 2). Age and marital age represent biological parameters of reproductive physiological maturation [61]. Older age and older age at marriage are associated with a higher probability of always-zero class membership, although, again, marital age was less influential at the oldest ages as indicated by the negative polynomial term.

The models in Table 2, taken together, suggest a fecundity threshold where Black women, compared to White women, have a higher probability of always-zero class membership and higher parity if fecund (Hypothesis 1). The ZINB model best fits the parity distribution in our sample and identifies a latent class of women at high risk of infecundity. We incorporate women's probability of always-zero latent class membership, *Infecundity*, into our mortality analyses to account for zero-category heterogeneity in examining observed parity—post-reproductive mortality associations.

## All-cause mortality, full sample models

Table 3 reports age-stratified proportional hazards estimates of all-cause mortality. Full model findings are available in S3 Table. In Model 1, the observed parity HRs pertaining to 0, 1, 4, 5, and 6+ births are all statistically significant, relative to a 2-birth reference. In Model 2, the *Infecundity* HR is associated with higher mortality and HRs for observed parities 1, 4, 5, and 6+ remain statistically significant; the 95% CI of the observed 0 birth category includes 1. Model 3 adjusts for birthplace region, IMR, childhood self-rated health, and family-of-origin socioeconomic variables; the *Infecundity* HR is statistically significant and approximately the same observed parity pattern holds, although the 4-birth category is only marginally significant. With the inclusion of post-reproductive-age adult health, marital, and socioeconomic statuses, parity—post-reproductive mortality associations are not statistically significant as a group. Consequently, we find only partial support for Hypothesis 2 although, notably, *Infecundity* remains significant in Model 4.

Parous women with one birth and with 6+ births (Model 5) have significantly elevated mortality risk relative to the 2-birth reference group, adjusting for timing variables. Observed parity indicators do not reach statistical significance as a group in Models 6 and 7 after adjusting for childhood and adulthood statuses, respectively. Regarding timing variables, an early age of first birth is statistically significant in Model 5, and in Model 6 with adjustment for childhood variables, but is not statistically significant in Model 7. However, in partial support for Hypotheses 3, the positive association between premarital birth and mortality remains statistically significant, adjusting for childhood and adulthood statuses (Models 5–7).

Overall, mortality risk is significantly elevated, with and without adjustment, among women with higher probabilities of *Infecundity* and among Black women. The parity—post-reproductive mortality relationship for all women (Models 1–3) and parous women (Model 5) exhibits a U-shaped distribution prior to adjustment for resources and statuses measured among survivors at post-reproductive ages.

## All-cause mortality, race/ethnic-specific models

Preliminary analyses revealed evidence of age*race/ethnicity interactions (see S2 Table). Thus, we examine parity and timing associations with mortality in race/ethnic-specific subsamples. Tables 4 and 5 present the results of age-stratified proportional hazards models for Black and

**Table 3. Stratified proportional hazards models: All-cause mortality, black and white women.**

| | All Women | | | | Parous Women | | |
|---|---|---|---|---|---|---|---|
| | Model 1 | Model 2 | Model 3 | Model 4 | Model 5 | Model 6 | Model 7 |
| | HR [95%CI] | HR [95%CI] | HR [95%CI] | HR [95%CI] | HR [95%CI] | HR [95%CI] | HR [95%CI] |
| Black Women | | | 1.44*** | 1.25* | | 1.42*** | 1.22* |
| | | | [1.21–1.71] | [1.05–1.49] | | [1.18–1.70] | [1.01–1.45] |
| *Reproductive Timing* | | | | | | | |
| Early First Birth | | | | | 1.15*** | 1.14*** | 1.06 |
| | | | | | [1.08–1.24] | [1.05–1.22] | [0.98–1.14] |
| Late First Birth | | | | | 1.08 | 1.09 | 0.97 |
| | | | | | [0.78–1.50] | [0.78–1.52] | [0.69–1.35] |
| Premarital Birth | | | | | 1.11** | 1.17*** | 1.15*** |
| | | | | | [1.04–1.20] | [1.09–1.27] | [1.07–1.24] |
| *Children Born* | | | | | | | |
| Infecundity Probability | | 1.08*** | 1.07*** | 1.05** | | | |
| | | [1.04–1.11] | [1.03–1.10] | [1.02–1.08] | | | |
| Observed 0 Births | 1.19** | 1.11+ | 1.15* | 1.09 | | | |
| | [1.06–1.33] | [0.99–1.25] | [1.01–1.30] | [0.96–1.23] | | | |
| Observed 1 Birth | 1.21*** | 1.18** | 1.13* | 1.08 | 1.19** | 1.14* | 1.08 |
| | [1.08–1.36] | [1.05–1.32] | [1.00–1.28] | [0.95–1.22] | [1.06–1.34] | [1.00–1.29] | [0.95–1.22] |
| Observed 3 Births | 1.01 | 1.01 | 1.02 | 1.00 | 0.99 | 1.01 | 0.99 |
| | [0.92–1.11] | [0.93–1.11] | [0.93–1.13] | [0.91–1.10] | [0.91–1.09] | [0.92–1.11] | [0.90–1.09] |
| Observed 4 Births | 1.10* | 1.11* | 1.10+ | 1.05 | 1.06 | 1.06 | 1.02 |
| | [1.00–1.22] | [1.00–1.22] | [1.00–1.22] | [0.94–1.16] | [0.96–1.17] | [0.96–1.18] | [0.92–1.14] |
| Observed 5 Births | 1.16* | 1.16* | 1.16* | 1.04 | 1.12+ | 1.11 | 1.01 |
| | [1.03–1.31] | [1.03–1.31] | [1.03–1.32] | [0.91–1.17] | [0.99–1.26] | [0.98–1.26] | [0.89–1.14] |
| Observed 6+ Births | 1.28*** | 1.28*** | 1.22*** | 1.12+ | 1.20*** | 1.15* | 1.09 |
| [Reference = 2 Births] | [1.15–1.42] | [1.15–1.42] | [1.08–1.36] | [1.00–1.26] | [1.08–1.34] | [1.02–1.29] | [0.97–1.23] |
| | | | | | | | |
| Parity U-Shape | Y | Y | Y | N | Y | Y | N |
| Parity χ2 Group Significance | *** | *** | ** | NS | *** | NS | NS |
| Model Wald Sandwich/df/N | 35.1/6/7322 | 55.6/7/7322 | 480.4/15/7322 | 1040.0/24/7322 | 71.0/8/6667 | 471.5/16/6667 | 1007.4/25/6667 |

Note: All models use cluster robust sandwich standard errors. Models 3, 6 adjust for birthplace, childhood health, family SES; Models 4, 7 further adjust for adult SES, health behaviors, marital, health status. Full models are available in S3 Table.

+ $p < .10$

* $p < .05$

** $p < .01$

*** $p < .001$ (two-tailed tests).

White women, respectively. In Models 1–4 of Table 4 (Black parous and nulliparous women), observed parity categories do not reach statistical significance as a group, although Models 1 and 2 suggest a U-shaped pattern in the observed parity distribution at the extremes of observed zero and 6+ children. In Models 2 through 4, *Infecundity* is significantly associated with elevated mortality risk. However, against expectations (Hypothesis 3), timing variables in Models 5–7 (Black parous women) do not reach statistical significance.

In Table 5 Models 1–3 (White parous and nulliparous women), observed parity variables are statistically significant as a group and exhibit a U-shaped distribution. In Model 3, HRs for 0, 1, 4, 5, and 6+ births are statistically significant, adjusting for *Infecundity*, birthplace, and

**Table 4. Stratified proportional hazards models: All-cause mortality, black women.**

| | All Women | | | | Parous Women | | |
|---|---|---|---|---|---|---|---|
| | Model 1 | Model 2 | Model 1 | Model 2 | Model 1 | Model 2 | Model 1 |
| | HR [95%CI] | HR [95%CI] | HR [95%CI] | HR [95%CI] | HR [95%CI] | HR [95%CI] | HR [95%CI] |
| *Reproductive Timing* | | | | | | | |
| Early First Birth | | | | | 1.04 | 1.13 | 1.07 |
| | | | | | [0.88–1.24] | [0.94–1.35] | [0.89–1.29] |
| Late First Birth | | | | | 0.92 | 0.92 | 0.77 |
| | | | | | [0.46–1.84] | [0.44–1.95] | [0.34–1.74] |
| Premarital Birth | | | | | 1.04 | 1.03 | 1.00 |
| | | | | | [0.89–1.22] | [0.87–1.22] | [0.84–1.17] |
| *Children Born* | | | | | | | |
| Infecundity Probability | | 1.06* | 1.08** | 1.06* | | | |
| | | [1.01–1.11] | [1.02–1.13] | [1.01–1.11] | | | |
| Observed 0 Births | 1.34* | 1.28+ | 1.14 | 1.16 | | | |
| | [1.02–1.78] | [0.97–1.70] | [0.83–1.55] | [0.84–1.58] | | | |
| Observed 1 Birth | 1.03 | 1.01 | 0.95 | 0.94 | 1.03 | 0.97 | 0.94 |
| | [0.78–1.36] | [0.76–1.33] | [0.70–1.27] | [0.70–1.25] | [0.77–1.36] | [0.72–1.31] | [0.70–1.27] |
| Observed 3 Births | 0.97 | 0.96 | 0.90 | 0.95 | 0.97 | 0.91 | 0.93 |
| | [0.75–1.25] | [0.74–1.25] | [0.69–1.19] | [0.72–1.25] | [0.75–1.25] | [0.68–1.20] | [0.71–1.23] |
| Observed 4 Births | 1.04 | 1.06 | 0.97 | 0.96 | 1.04 | 0.95 | 0.95 |
| | [0.79–1.38] | [0.80–1.40] | [0.73–1.30] | [0.72–1.27] | [0.78–1.37] | [0.71–1.27] | [0.71–1.26] |
| Observed 5 Births | 1.06 | 1.08 | 1.07 | 0.98 | 1.05 | 1.02 | 0.95 |
| | [0.79–1.41] | [0.81–1.44] | [0.80–1.43] | [0.72–1.32] | [0.78–1.40] | [0.76–1.36] | [0.70–1.29] |
| Observed 6+ Births | 1.23+ | 1.27* | 1.20 | 1.06 | 1.21 | 1.12 | 1.05 |
| | [0.98–1.55] | [1.01–1.60] | [0.94–1.53] | [0.83–1.36] | [0.96–1.53] | [0.87–1.44] | [0.81–1.35] |
| Parity U-Shape | Y | Y | N | N | Y | N | N |
| Parity χ2 Group Significance | NS | NS | NS | NS | NS | NS | NS |
| Model Wald Sandwich/ df/N | 10.1/6/1274 | 15.0/7/1274 | 102.8/13/1274 | 218.4/22/1274 | 6.8/8/1140 | 99.2/14/1140 | 216.2/23/1140 |

Note: All models use cluster robust sandwich standard errors. Models 3, 6 adjust for birthplace, childhood health, family SES; Models 4, 7 further adjust for adult SES, health behaviors, marital, health status. Full models are available in S4 Table.

+ $p < .10$

* $p < .05$

** $p < .01$

*** $p < .001$ (two-tailed tests).

early-life-course selection factors. In Model 4, observed parity HRs do not reach statistical significance as a group with adjustment for adulthood variables. In Models 5 and 6 (White parous women), an early age of first birth, premarital birth, and having one-birth, relative to a 2-birth reference group, are associated with higher mortality, although the observed parity variables, as a group, are only statistically significant in Model 5. Full Model 7 finds partial support for Hypothesis 3: only premarital timing retains statistical significance.

Fig 1 illustrates Black:White observed parity—post-reproductive mortality associations taken from Tables 4 and 5. For White women (Panel A), the unadjusted Model and the Model that adjusts for *Infecundity* risk only are U-shaped; the *Infecundity* risk adjustment only reduces the observed zero parity HR, while other observed parity HRs overlap the unadjusted HRs. This suggests that *Infecundity* primarily differentiates mortality risk among the childless. Further adjustment for childhood and adulthood variables in the fully adjusted model flattens the U-shape; parity categories are not statistically significant as a group (Table 5).

**Table 5. Stratified proportional hazards models: All-cause mortality, white women.**

| | *All Women* | | | | *Parous Women* | | |
|---|---|---|---|---|---|---|---|
| | **Model 1** | **Model 2** | **Model 3** | **Model 4** | **Model 5** | **Model 6** | **Model 7** |
| | **HR [95%CI]** | **HR [95%CI]** | **HR [95%CI]** | **HR [95%CI]** | **HR [95%CI]** | **HR [95%CI]** | **HR [95%CI]** |
| *Reproductive Timing* | | | | | | | |
| Early First Birth | | | | | 1.17*** | 1.13** | 1.05 |
| | | | | | [1.08–1.26] | [1.04–1.23] | [0.97–1.14] |
| Late First Birth | | | | | 1.12 | 1.15 | 1.05 |
| | | | | | [0.80–1.57] | [0.80–1.64] | [0.73–1.49] |
| Premarital Birth | | | | | 1.11** | 1.21*** | 1.19*** |
| | | | | | [1.03–1.20] | [1.12–1.32] | [1.10–1.30] |
| *Children Born* | | | | | | | |
| Infecundity Probability | | 1.07** | 1.07** | 1.06* | | | |
| | | [1.02–1.12] | [1.02–1.12] | [1.01–1.11] | | | |
| Observed 0 Births | 1.13* | 1.07 | 1.14* | 1.10 | | | |
| | [1.00–1.28] | [0.94–1.22] | [1.00–1.30] | [0.92–1.20] | | | |
| Observed 1 Birth | 1.24*** | 1.22** | 1.18* | 1.10 | 1.22** | 1.17* | 1.10 |
| | [1.09–1.40] | [1.08–1.38] | [1.02–1.35] | [0.96–1.26] | [1.08–1.38] | [1.02–1.34] | [0.96–1.26] |
| Observed 3 Births | 1.01 | 1.02 | 1.04 | 1.00 | 0.99 | 1.02 | 1.00 |
| | [0.92–1.12] | [0.92–1.12] | [0.94–1.15] | [0.91–1.11] | [0.90–1.10] | [0.92–1.12] | [0.90–1.10] |
| Observed 4 Births | 1.10+ | 1.11+ | 1.12* | 1.05 | 1.06 | 1.08 | 1.03 |
| | [0.99–1.22] | [1.00–1.23] | [1.01–1.25} | [0.94–1.18] | [0.95–1.18] | [0.96–1.20] | [0.92–1.15] |
| Observed 5 Births | 1.16* | 1.16* | 1.18* | 1.03 | 1.11 | 1.13+ | 1.01 |
| | [1.01–1.32] | [1.01–1.33] | [1.03–1.36] | [0.90–1.19] | [0.97–1.27] | [0.98–1.30] | [0.88–1.16] |
| Observed 6+ Births | 1.18* | 1.18** | 1.19** | 1.12 | 1.12+ | 1.13+ | 1.10 |
| | [1.04–1.34] | [1.04–1.34] | [1.04–1.37] | [0.98–1.29] | [0.99–1.28] | [0.99–1.30] | [0.96–1.26] |
| Parity U-Shape | Y | Y | Y | N | Y | Y | N |
| Parity χ2 Group Significance | ** | ** | * | NS | * | NS | NS |
| Model Wald Sandwich/ df/N | 19.3/6/6048 | 27.2/7/6048 | 338.8/13/6048 | 794.3/22/6048 | 53.8/8/5527 | 347.2/14/5527 | 792.8/23/5527 |

Note: All models use cluster robust sandwich standard errors. Models 3, 6 adjust for birthplace, childhood health, family SES; Models 4, 7 further adjust for adult SES, health behaviors, marital, health status. Full models are available in S5 Table.

+ p < .10

* p < .05

** p < .01

*** p<. 001 (two-tailed tests).

## Discussion

This study examined the parity—post-reproductive mortality relationship and race/ethnic-related differences in this relationship attributable to women's differential biophysiological and social likelihoods of bearing children. We distinguished a nexus of biophysiological and social-environmental forces by focusing on the childbearing contexts of historical birth cohorts and used count regression methods to estimate women's expected *Number of Children Born*, as well as their probability of membership in a latent always-zero parity class (*Infecundity*). As the ZINB equation best fit the data (Table 2), a first main finding is that infecundity was non-random and identifiable among Black and White women in the HRS sample that we studied. Considering both portions of the ZINB equation together, Black women had a higher mean *Number of Children Born* and higher *Infecundity* risk than White women. These results provide evidence for Hypothesis 1.

**(A)**

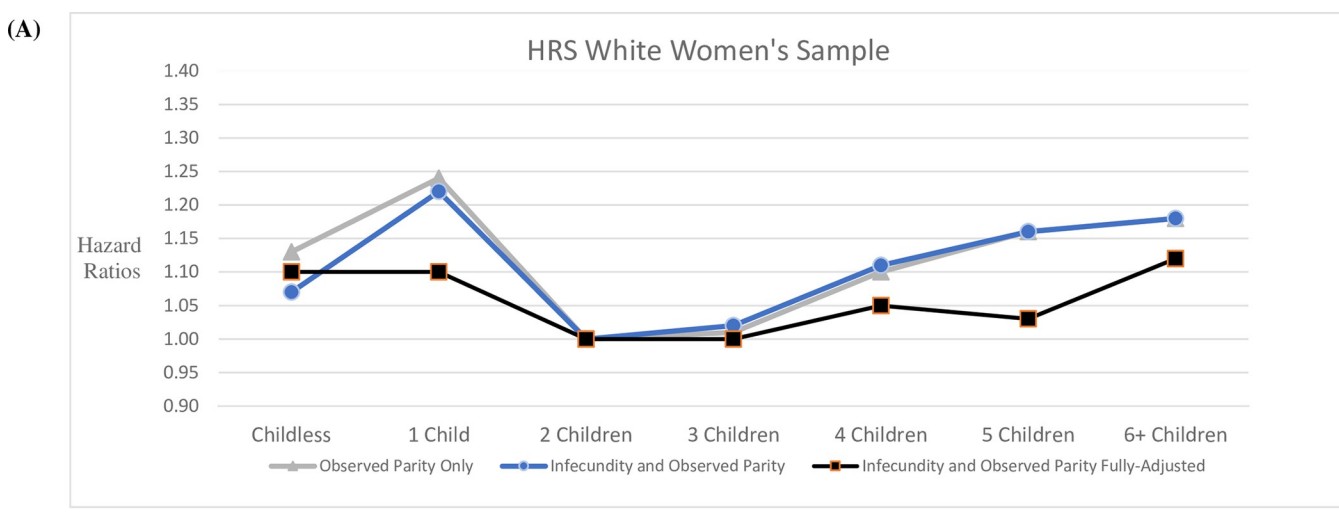

**(B)**

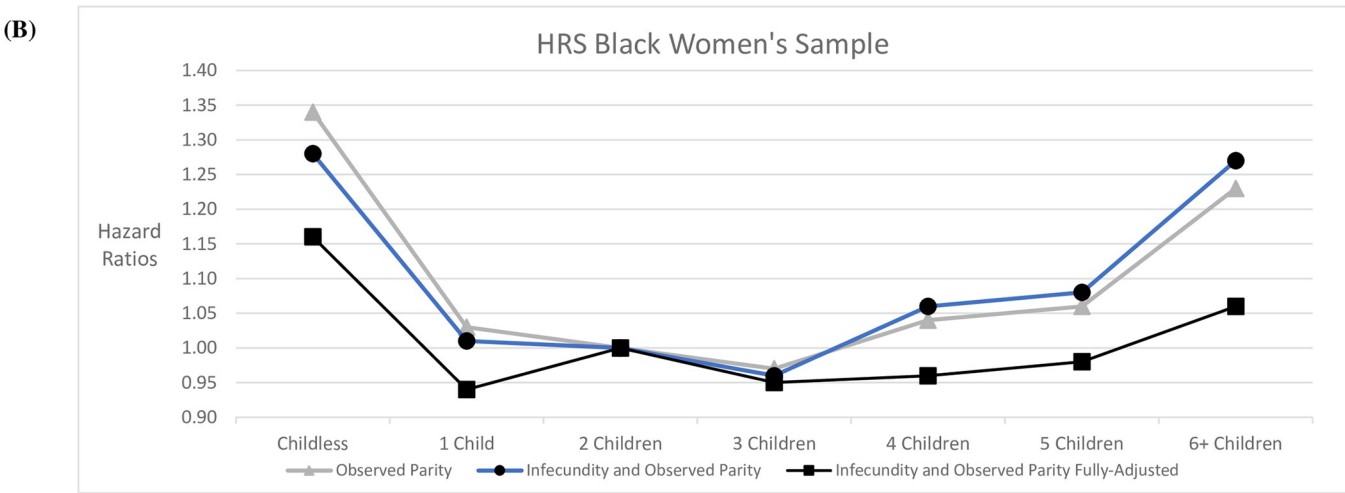

**Fig 1. Observed parity-mortality hazard ratios, with and without model adjustments.** For Black women (Panel B), in Models unadjusted and only adjusted for *Infecundity* risk, U-shapes suggest elevated mortality risk at parities 0 and 6+; with adjustment for *Infecundity* risk, the HRs decrease for observed parities zero and one and increase for observed parities 4, 5, and 6+. However, the observed parity categories for Black women are not statistically significant as a group (Table 4).

We next estimated age-stratified proportional hazards models to test the associations between *Infecundity* risk, observed parity, and mortality in a pooled-sample of Black and White women (Table 3) and in race/ethnic-specific samples of Black and White women (Tables 4 and 5, respectively). Our second main finding is that a higher risk of *Infecundity* was always statistically associated with higher mortality risk for Black and White women, in pooled and race/ethnic-specific models. No adjustments to these models fully attenuated the mortality risk associated with higher *Infecundity* risk. In addition, observed parity—post-reproductive mortality associations in Tables for all women, Black women, and White women were U-shaped in models that did and did not adjust for *Infecundity risk;* this third finding provides partial support for Hypothesis 2.

Notably, marked race-ethnic differences also exist in observed parity—post-reproductive mortality associations. The associations did not reach statistical significance, as a group, in Black women's models (Table 4). By contrast, in the sample of White women, a statistically significant U-shaped distribution of parity—post-reproductive mortality associations (Table 5),

relative to a 2-birth reference, was present in models that did and did not adjust for *Infecundity* risk and early-life-course measures (i.e., Models 1, 2, and 3). Only the adjustment for resources and statuses, measured at post-reproductive ages, reduced observed parity—post-reproductive mortality associations, as a group, to non-significance (Model 4). Thus, a fourth main finding is that parity—post-reproductive mortality relationships differed by race/ethnicity in the birth cohorts we studied.

Why might this be so? Parity–mortality studies generally elide childbearing-context differences involving population fertility control practices [26] and survival environments [6]. Most U.S. Black women in pre-1940s birth cohorts—85% in the HRS sample—were southern-born, hence exposed to Jim Crow-era survival environments where "race," a socio-legal organizing principle, allocated access to schools, health care, and occupations other than agriculture [48,49,50]. In our study, Black women exhibited a parity—post-reproductive mortality pattern (Table 4, Model 1) resembling that found in other historical birth cohorts, of higher mortality among the childless [26,27,28]. Further differentiation of "zero births" (Model 2) revealed that the higher mortality was associated with *Infecundity* (poor reproductive health). Black women in pre-1940s birth cohorts had high fertility as well (Table 2) because the southern racialized economy tied their children's economic value to agricultural field work more than education [23,33,48]. Black women's poorer reproductive health and higher fertility was further linked to their children's poorer survival odds [23] as high child mortality rates can foster precautionary avoidance of fertility control [66]. But, while Black women bore more children than White women, and their parity categories formed a U-shaped pattern (Fig 1), the categories were not statistically significant as a group (Table 4). We speculate that, because some infecundity (inability to reproduce) might have occurred after childbearing commenced (after a live birth) [16], Black women—due to poorer reproductive health—likely experienced infecundity risk with each birth, reducing their likelihood of reaching higher parity while suppressing the high-parity statistical risk of mortality. In the U.S. context, Black women would be more subject to "healthy pregnant women" selectivity, where healthier women were fecund, gave birth, and—because higher fertility amidst poor health and/or less repletion still carries its own health risks—survived to have a next child.

In contrast, White women's U-shaped parity—post-reproductive mortality patterns, prior to adjustment for adult resources, resemble those found in contemporary birth cohorts (Table 5, Models 1–3) [25,26,27]. More striking, White women who had survived to post-reproductive ages had a low-parity peak in mortality risk at one child. This suggests that, subject to comparatively better survival environments over childbearing years, White women more likely achieved a first birth in spite of lower overall fertility (Table 2). Their expectations of better child survival odds could promote fertility control practices. Additionally, White women's better post-reproductive survival environments (e.g., higher living standards) statistically reduced both lower- and higher-parity post-reproductive mortality risk (Table 5, Model 4), net of *Infecundity*, although *Infecundity* remained significantly associated with post-reproductive mortality.

Population-level studies of early twentieth-century southern Black women link their poor reproductive health to extremely high rates of disease, poverty, and lack of medical care [17,46,67]. In the analyses based on the pooled sample (Table 3), we find that nonparous and parous Black women have a 25% increased risk of mortality (Model 4) and parous Black women have a 22% increased risk of mortality (Model 7). Hence, a fifth finding is that the association of Black ancestry with post-reproductive mortality is not fully explained, despite adjustments for: *Infecundity*; observed parity; and health and socioeconomic selectivity in childhood and at study enrollment in adulthood (Hypothesis 4). The persistent significance of race/ethnicity in fully adjusted models suggests additional, yet-unmeasured structural forces

contributed to Black women's higher rates of post-reproductive mortality, including cumulative exposure to the Jim Crow-era policies marked by racial discrimination, residential segregation (geographic place), and limited access to health-supportive social services and institutions (e.g., hospitals, institutions of higher education) [24].

Our study benefits from linking the RAND-harmonized HRS files to many other HRS-related databases. As a result of these linkages, we were able to examine women in the same historical birth cohorts with different reproductive practices and differentiate their lives across a wide range of life-course statuses associated with fertility and mortality (e.g., childhood background, education, later-life home ownership status). The data facilitated our ability to capture early-life-course selectivity in the same historical birth cohorts at the "front end" of parity-related mortality risk—i.e., the higher mortality risk associated with nulliparity and very low parity. Additionally, the prospective nature of the RAND allowed us to track participants for an average of 18.6 years, which is longer than most other U.S. parity—post-reproductive mortality studies. Combining the RAND with Vital Statistics data allowed us to examine: birth-place IMR, associated with community living standards and public health policies; socioeconomic selectivity early and late in life; and marital resources and health at survey baseline.

However, despite its strengths, this study cannot track selective processes influencing mortality that might have occurred during women's reproductive years. Because of the age-eligibility criteria for enrollment in the HRS, our findings are conditional on women's survival to midlife. Among fecund women, survival of a pregnancy enables progression to a higher parity category. This selective process evokes a default "healthy pregnant women effect," which is most visible in high-mortality, high-fertility populations [25]. We know that early twentieth-century reproductive-age mortality was higher for Black than White women in the birth cohorts included in this study [64]. Consequently, our models likely under-estimate mortality risk among parous Black women. Other indirect evidence of reproductive period selection involving Black women is that their expected number of children is 40% higher than among White women—if they were fecund. Of course, having additional children required surviving prior pregnancies, which similarly evokes the default "healthy pregnant women effect" [25].

While our models contribute to understanding of the parity—post-reproductive mortality relationship by measuring selectivity in the transition to having at least one child (i.e., *Infecundity*), due to the design of the HRS, we are unable to take selectivity in the survival of pregnancies among parous women into account. We cannot directly measure the mortality risk that surrounds having an additional child because we only observe women during the post-reproductive period. Disposable soma [12,21,25] and maternal repletion [44] scholars warn that examining current or completed parity in a sample of women can mask the parity-mortality relationship; information about infecundity and maternal mortality risk surrounding parity progression is omitted. To better understand the parity-mortality relationship, future data collection and research should concentrate on the formative influences of early-life exposures on women's reproductive health and parity progression in reproductive careers. Studies of weathering processes that more fully measure the social determinants of health are needed.

Despite these limitations, our findings address an important gap in this research area. It is common practice in U.S. (and most extant) studies to include an observed birth category of zero births, or omit the category entirely and examine parous women only. However, as we show, women faced different contexts of selection into childlessness and their probability of selection affected their later mortality risk, net of other variables.

Much about the parity—post-reproductive mortality relationship remains unknown. The results of this study suggest directions for future research. For example, we control for number of medical conditions in adulthood in the models of all-cause mortality. However, no study to

date has addressed the specific, prospective dynamics at the nexus of infecundity, parity, and the likelihoods (pathways) of individual medical conditions, their cumulation by later-life, and mortality. Clarifying these relationships will require better data collection to allow for more detailed consideration of the etiologies of these conditions and their associations with women's parity and pregnancy histories, as well as their associations with mortality. Clarifying the relationships between infecundity, parity, and the likelihood, onset timing, and/or prevalence of individual medical conditions, and the pace of their accumulation, is especially important considering Black American women's earlier trajectory of health decline [11,50].

Continued research on the social and biophysiological determinants of infecundity is warranted. Additionally, the onset of health deterioration earlier in the life course more directly overlaps the reproductive years of Black than White women and reflects—among current as well as historical birth cohorts—lack of equitable access to societal resources, including health care [68]. The results of this study indicate directions for ameliorative health policies, including the need to institutionalize universal, maternal care services that provide an added fourth trimester of post-delivery care. Past studies stressing maternal depletion, and current studies that locate most maternal mortality among today's Black women in the post-partum period, all point to the need for post-delivery care to detect and remediate reproductive health problems [68]. Additionally, greater translational application of current weathering/reproduction research is warranted. In addition to a need for universal, expanded reproductive care, there is need to restructure it, with less segregation/siloing of care outside of primary care in order to optimize maternal health prior to a first pregnancy and provide on-going monitoring of longer-run sequelae of pregnancy [69].

Finally, our results highlight the importance of biophysiological and social environmental factors and how they combine to shape the parity—mortality relationship. Our results emphasize that the interweaving of diverse factors produce different fertility processes, reproductive outcomes, and differential mortality risk, also associated with race/ethnicity, in a manner indicating health disparities. It is important that future work examine race/ethnicity-related reproductive health disparities and the historical childbearing contexts that generate them.

## Supporting information

**S1 Table. Sample construction and data flow.**
(PDF)

**S2 Table. Full age-adjusted proportional hazards models: All-cause mortality, black and white women.**
(PDF)

**S3 Table. Full age-stratified proportional hazards models: All-cause mortality, black and white women.**
(PDF)

**S4 Table. Full age-stratified proportional hazards models: All-cause mortality, black women.**
(PDF)

**S5 Table. Full age-stratified proportional hazards models: All-cause mortality, white women.**
(PDF)

**S6 Table. Reproductive timing sensitivity analysis: Stratified proportional hazards models: All-cause mortality, black and white women.**
(PDF)

## Author Contributions

**Conceptualization:** Cheryl Elman.

**Data curation:** Cheryl Elman.

**Formal analysis:** Cheryl Elman, Angela M. O'Rand, Andrew S. London.

**Funding acquisition:** Angela M. O'Rand.

**Investigation:** Andrew S. London.

**Methodology:** Cheryl Elman, Angela M. O'Rand, Andrew S. London.

**Project administration:** Cheryl Elman.

**Visualization:** Angela M. O'Rand.

**Writing – original draft:** Cheryl Elman, Angela M. O'Rand, Andrew S. London.

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
