## [Decision Letter · Decision Letter 0]

3 Jun 2024

PONE-D-23-43696Mid-Twentieth Century Black-White disparities in U.S. women’s reproductive health and later- life mortality: Implications for contemporary health research and policyPLOS ONE

Dear Dr. Elman,

Thank you for submitting your manuscript to PLOS ONE. After careful consideration, we feel that it has merit but does not fully meet PLOS ONE’s publication criteria as it currently stands. Therefore, we invite you to submit a revised version of the manuscript that addresses the points raised during the review process.

The reviewers were enthusiastic about the significance of the work but have identified some areas where there could be more clarity, especially when considering the likely multidisciplinary audience.

We look forward to receiving your revised manuscript.

Kind regards,

Emily W. Harville

Academic Editor

PLOS ONE

2. Please provide additional details regarding ethical approval in the body of your manuscript. In the Methods section, please ensure that you have specified the name of the IRB/ethics committee that approved your study.

 [The first two authors, C. Elman and A. O'Rand, received research project support from an NICHD Population Dynamics Research Infrastructure Program award to the Duke Population Research Center (P2C HD065563) and an NIA Centers on the Demography and Economics of Aging Program award to the Duke Center for Population Health and Aging (P30 AG034424) at the Duke Population Research Institute. The third author, A. London, received research project support from an NIA Centers of the Demography and Economics of Aging award to the Center for Aging and Policy Studies in the Aging Studies Institute at Syracuse University (P30 AG066583).].  

5. We note that you have indicated that there are restrictions to data sharing for this study. PLOS only allows data to be available upon request if there are legal or ethical restrictions on sharing data publicly. For more information on unacceptable data access restrictions, please see http://journals.plos.org/plosone/s/data-availability#loc-unacceptable-data-access-restrictions. 

Reviewers' comments:

Reviewer's Responses to Questions

**Comments to the Author**

1. Is the manuscript technically sound, and do the data support the conclusions?

Reviewer #1: Yes

Reviewer #2: Partly

2. Has the statistical analysis been performed appropriately and rigorously? 

Reviewer #1: Yes

Reviewer #2: I Don't Know

3. Have the authors made all data underlying the findings in their manuscript fully available?

Reviewer #1: No

Reviewer #2: Yes

4. Is the manuscript presented in an intelligible fashion and written in standard English?

Reviewer #1: Yes

Reviewer #2: Yes

5. Review Comments to the Author

Reviewer #1: This is a well-done paper that adds to the growing evidence on women’s mortality risk associated with parity and infecundity. I genuinely enjoyed reading this manuscript - the authors skillfully situate their study within the broader context of existing evidence, demonstrating a comprehensive understanding of the relevant literature. The analyses presented are methodologically sound and rigorous, and the findings offer valuable insights into understanding the complex dynamics of underlying reproductive health disparities by race/ethnicity.

I commend the authors for their outstanding work and only have very minor suggestions that focus on clarifications to further strengthen the manuscript's coherence:

- In the abstract and throughout the paper, the authors claim to examine the infecundity risk and parity by race – my suggestion would be to clarify that it is “by race/ethnicity” as they analyze the sample with non-Hispanic Black and White women (i.e., the combined race and ethnicity variable is used and therefore the term race/ethnicity is more accurate to signify the combination of both race and ethnicity into a single variable).

- The authors skillfully place their work in the context of previous evidence and present three aims addressed in their paper (p.11). I would like to see more specific hypotheses outlined as well (regarding the main associations and any moderation effects examined).

- Under “Adjustment Measures” (p.16) the authors explain that they assigned the never-married women the mean “Age at Marriage” instead of excluding them. What is the proportion of “never married” women in the sample? It is not clear why creating a categorical variable which would allow to more accurately capturing categories of age at marriage (e.g., never married, married before 20; etc.) was not chosen instead?

- In Table 1, please consider including the p-value of the difference between the two racial/ethnic groups.

- In Table 2, please include the 95% CI next to reporting odds ratios.

- The Discussion section is very well-written with the main findings outlined and discussed in sufficient detail. However, I would suggest enhancing the “implications for contemporary health research and policy” component indicated in the title of this paper. More specifically, based on the main findings of this study, what are the key policy implications in addressing racial/ethnic inequities in reproductive health in the US?

Reviewer #2: This is an interesting paper which uses a two-step analytic process, first to separate voluntary from involuntary childlessness and therefore to estimate a risk of infecundity (involuntary childlessness) for individual women. The next step estimates the effect of this infecundity risk and of parity on later life mortality, and compares the results for black and white women in the US. The authors conclude that black women’s later life mortality risk is associated with their infecundity. The analyses seem appropriate (although I’m not an expert in these precise models and feel they could be better explained for the audience) and imaginative.

Literature review and framing.

However I think there are improvements the authors could make to the framing of the paper in order to make it more accessible. In particular, the focus on childlessness and post-reproductive mortality emerges rather slowly during the introduction and literature review. This could be made much clearer from the start and the reader would benefit from greater clarity in the literature review too.

I also feel that the paper would benefit from closer attention to the historical context that the study cohorts were going through. In particular, I would have liked some recognition that the cohorts of women studied (born 1920-1941) were the cohorts going through childbearing during the baby boom. We know that such women had unusual marriage and childbearing patterns compared to those before and after. How might this affect conclusions? Were there differences in the experience of the baby boom that affected black and white women differently? It might be easier to conceptualise these baby boom generations as an atypical episode rather than in between two ‘cross-overs’ (lines 123-126).

It would also be good to recognise (eg line 71) that historically the most important driver of childnessness was celibacy (non-marriage). Fecund women were likely to have been selected into marriage through pre-marital pregnancy.

Sometimes an awful lot of work is done by the term ‘infecundity’ (eg Line 130: the use of ‘infecundity’ here seems to represent a composite of everything excluding voluntary control, but this includes a wide range of factors). How can you tell the differences between infecundity (as in the biological incapacity to conceive), exposure (eg through marriage/partnership), and voluntary childlessness (use of contraception)? Demographers have specific meaning for fecundity and fertility, but these terms are not used in the same way by every field or by the general public. Clarity in how the authors are using them would be helpful. This is also relevant for lines 169-171: Because most measures of fertility include only live births, the ‘healthy pregnant woman effect’ – surely this selection is not just about healthy pregnant women, but healthy women who produce live births. Similarly on line 191, ‘the physiological resilience to the fecund’ should be ‘the physiological resilience to be fertile’.

I’m surprised that the proximate determinants of fertility framework is not mentioned in the text (Bongaarts is referenced in the literature) - this might help with the conceptualisation.

In ‘Biophysiological factors’ I’m surprised that the protective effect of breastfeeding on the risk of breast cancer was not mentioned.

Line 111: ‘natural fertility’ should be defined for those who are not familiar with the term

In the ‘Socio-economic factors’ section (lines 175-228), some of the factors mentioned explain the increased risks associated with low parities, and some explain the increased risk of high parities: it would be helpful to the reader if these were distinguished.

Line 260: ‘The RAND documents question changes and …’ I had to read this several times before I saw documents as the verb ad question as a noun rather than the other way around. I suggest a rephrase.

Data.

Regarding the observed parity and timing indicators (lines 301-311), age at first birth is calculated using the age of participant and the age of their first child. This implies that respondents were asked for the ages of all their children, including the age that any who had died would be had they lived. This might be subject to more error than the ages of living children, and it would be good to have an assessment of the effect of this on results.

Treatment of missing data: did you do sensitivity analyses to examine how much the use of mother’s education when father’s was missing?

Analytic strategy.

I would have appreciated more explanation of how the combination of the logistic and count processes can distinguish voluntary and involuntary childlessness. Does this depend on an assumption that you can predict the proportion of women who are voluntarily childless from the subsequent parity distribution. Why would this be so? How do these models differ from ‘cure’ models?

Lines 401-403: explain the significance of this correlation.

Results.

Fertility models:

Age and cohort. Age (eg on line 448) presumably means age at survey, but which round of the survey? It might be less confusing to present this as cohort. If it is not cohort, what is its relevance? On line 470-1 you say that ‘age and marital age represent biological parameters of reproductive physiological maturation’ – I’m not sure how age at survey would do that, or does this now refer to age at birth?

Results are conditional on surviving to the survey date: how might differential survival up to that point affect results?

Descriptive information (Table 1) indicates that the parents of blacks were better educated than the parents of whites, but the black respondents themselves were worse educated. This seems unexpected – is there some sort of strange selection operating?

Count model: presumably this refers just to women who are not infecund/infertile? This should be made clearer.

More guidance is needed on how to interpret the models, for example the beta coefficient in Table 2. Why not also give the exponentiated values in the count bit of the Table (particularly as these are reported in the text). In the ‘always-zero’ bit of the Table, could the odds ratio be shown?

6. PLOS authors have the option to publish the peer review history of their article (what does this mean?). If published, this will include your full peer review and any attached files.

Reviewer #1: No

Reviewer #2: No

---

## [Author Response · Author response to Decision Letter 0]

24 Jul 2024

To the Editor and Reviewers:

We are grateful for the careful attention that both reviewers paid to our paper and for the chance to revise it. In response to the reviewers’ comments, we streamlined the front-end of the paper and updated the Tables presented in the Results section. Additionally, in response to Reviewer 1’s comments, we changed all references to “race” to “race/ethnicity” or “race/ethnic,” made our hypotheses more explicit, and enhanced our discussion of the health policy and research implications of our study. In response to Reviewer 2’s comments, we better defined and differentiated infecundity and fertility, added discussion of anomalous childbearing and marital behaviors of 1920-1941 birth cohorts associated with the historical baby boom generation, revised the description of count regression methods for the analytic strategy section, and responded to specific variable coding issues.

Below, we provide detailed responses to each of the Reviewers’ comments. We believe that we have responded to all of their comments and that our paper is stronger as a result. We are hopeful that our revised paper will be deemed suitable for publication in PLOS One. If additional changes are required prior to acceptance, we would welcome the opportunity to make them. 

All references below are included in the bibliography of the paper.

Reviewer #1 

This is a well-done paper that adds to the growing evidence on women’s mortality risk associated with parity and infecundity. I genuinely enjoyed reading this manuscript - the authors skillfully situate their study within the broader context of existing evidence, demonstrating a comprehensive understanding of the relevant literature. The analyses presented are methodologically sound and rigorous, and the findings offer valuable insights into understanding the complex dynamics of underlying reproductive health disparities by race/ethnicity. I commend the authors for their outstanding work and only have very minor suggestions that focus on clarifications to further strengthen the manuscript's coherence.

Response: We are grateful that Reviewer 1 thinks our study is well-done and enjoyed reading it. We very much appreciate Reviewer 1’s thoughtful comments and suggestions. We have considered all of them carefully as we have revised our manuscript. 

In the abstract and throughout the paper, the authors claim to examine the infecundity risk and parity by race – my suggestion would be to clarify that it is “by race/ethnicity” as they analyze the sample with non-Hispanic Black and White women (i.e., the combined race and ethnicity variable is used and therefore the term race/ethnicity is more accurate to signify the combination of both race and ethnicity into a single variable).

Response: Thank you for this suggestion. 

Changes we made: We replaced the term “race” with “race/ethnicity” or “race/ethnic” throughout the paper.

The authors skillfully place their work in the context of previous evidence and present three aims addressed in their paper (p.11). I would like to see more specific hypotheses outlined as well (regarding the main associations and any moderation effects examined).

Response: We appreciate Reviewer 1’s helpful comment. 

Changes we made: In response, we made our hypotheses more explicit. Specifically, we added four hypotheses. We now list them at the end of the Specific Aims section of the paper (Lines 258-278), and refer to them in the Results and Discussion sections of the paper. The four hypotheses we added to the paper are listed below.

Regarding Fertility:

Hypothesis 1: We expect to find that Black women have a higher probability of infecundity, but, at the same time, an equal or higher mean number of births relative to White women. 

Regarding Mortality:

Hypothesis 2: We expect to find a U-shaped parity-mortality association such that infecundity risk, low parity, and high parity, relative to 2 births, are positively associated with mortality. 

Hypothesis 3: We expect to find that the life-course timing of childbearing is associated with mortality such that premarital birth, and younger and older age at first birth, elevate mortality risk. 

Hypothesis 4: We expect to find that race-related differences in mortality risk are only partially explained by parity and its timing, such that Black women will have greater post-reproductive mortality risk than White women. 

Under “Adjustment Measures” (p.16) the authors explain that they assigned the never-married women the mean “Age at Marriage” instead of excluding them. What is the proportion of “never married” women in the sample? It is not clear why creating a categorical variable which would allow to more accurately capturing categories of age at marriage (e.g., never married, married before 20; etc.) was not chosen instead?

Response: The number of never-married women in our sample is 212 (2.9%), as shown in Table 1. Their distribution is as follows:

 0 Births 1 Birth 2 Births 3 Births 4 Births 5 Births 6+ Births Total

Never Married

N 150 17 14 16 7 3 5 212

% 70.9 8.0 6.6 7.5 3.3 1.1 2.4 100 

Originally, we considered various approaches for managing this small group of never-married women but hesitated to exclude them (see below). We decided to include them by assigning them the mean age at marriage and including a never-married flag to be included in the model. In response to this comment, we did two things. We re-ran models using the categorical approach suggested by the reviewer and we re-ran all of the models in Tables 2-5 on the sample that excluded the 212 never-married women. In both cases, the substantive results were the same as those we report in the paper. The results are robust. Because the polynomial functional form is roughly equivalent to the categorical form and is the most-utilized functional form in demographic studies of fertility, we retained our original approach in the revised paper. It is our preferred approach. We hope our due diligence in response to this comment is satisfactory to Review 1. 

Changes we made: We now provide a citation to a study (Line 449, Guinnane et al. 2002) that uses a similar age at marriage polynomial approach. We also added at Lines 384-392 the following: 

Preliminary analysis of the joint distributions of Age at Marriage and Children Ever Born indicated that almost a third of never-married women had children and nearly 20% had two or more children. HRS/AHEAD women were not queried about non-marital unions, known to be associated with model variables of race/ethnicity, social class, health resources, and birth timing. Rather than exclude them, we assigned never-married women the mean Age at Marriage for the sample. flagging the substitution in the Children Ever Born count analyses. Sensitivity analyses that: (1) used age at marriage categories including never-married and (2) excluded all never-married women produced the same substantive results as those we report in the paper.

In Table 1, please consider including the p-value of the difference between the two racial/ethnic groups.

Change we made: We now include this. 

In Table 2, please include the 95% CI next to reporting odds ratios.

Change we made: We now include this.

The Discussion section is very well-written with the main findings outlined and discussed in sufficient detail. However, I would suggest enhancing the “implications for contemporary health research and policy” component indicated in the title of this paper. More specifically, based on the main findings of this study, what are the key policy implications in addressing racial/ethnic inequities in reproductive health in the US?

Change we made: We address this in the final paragraphs, as follows, between Lines 722 and 733: 

Continued research on the social and biophysiological determinants of infecundity is warranted. Additionally, the onset of health deterioration earlier in the life course more directly overlaps the reproductive years of Black than White women and reflects—among current as well as historical birth cohorts—lack of equitable access to societal resources, including health care.67 The results of this study indicate directions for ameliorative health policies, including the need to institutionalize universal, maternal care services that provide an added fourth trimester of post-delivery care. Past studies stressing maternal depletion, and current studies that locate most maternal mortality among today’s Black women in the post-partum period, all point to the need for post-delivery care to detect and remediate reproductive health problems.67 Additionally, greater translational application of current weathering/reproduction research is warranted. In addition to a need for universal, expanded reproductive care, there is need to restructure it, with less segregation/siloing of care outside of primary care in order to optimize maternal health prior to a first pregnancy and provide on-going monitoring of longer-run sequelae of pregnancy.68 

Reviewer #2: 

This is an interesting paper which uses a two-step analytic process, first to separate voluntary from involuntary childlessness and therefore to estimate a risk of infecundity (involuntary childlessness) for individual women. The next step estimates the effect of this infecundity risk and of parity on later life mortality, and compares the results for black and white women in the US. The authors conclude that black women’s later life mortality risk is associated with their infecundity. The analyses seem appropriate (although I’m not an expert in these precise models and feel they could be better explained for the audience) and imaginative.

We are pleased that Reviewer 2 found our paper interesting. We very much appreciate Reviewer 2’s thoughtful comments and suggestions. We have considered all of them carefully as we have revised our manuscript. 

Literature review and framing.

1. However, I think there are improvements the authors could make to the framing of the paper in order to make it more accessible. In particular, the focus on childlessness and post-reproductive mortality emerges rather slowly during the introduction and literature review. This could be made much clearer from the start and the reader would benefit from greater clarity in the literature review too.

Response: We thank you for this observation and have revised the paper based on these comments (described below). However, please let us clarify that our investigation of the childlessness—post-reproductive mortality association is only one element in our study. We are as interested in the overall relationship between parity and post-reproductive mortality. Put differently, we are concerned that extant research on the parity—post-reproductive mortality relationship in the U.S. appears anomalous with respect to functional form (U-shape, linear, no association) relative to findings from a large body of non-U.S. studies because heterogeneity in childlessness status and race-related differences have not been adequately addressed. Because a growing literature from other Western nations consistently demonstrates a U-shaped parity—post-reproductive mortality relationship and U.S. studies show no consistent pattern, our first Introductory paragraph integrates U.S. studies into the larger field of parity—post-reproductive mortality research. We consider this contextualization essential to our project and think that the discussion in this first paragraph is an appropriate start to our paper.

Changes we made. This said, in response to Reviewer 2’s suggestion, we modified the Introduction in several ways to try to elevate the childlessness—post-reproductive mortality component of the parity—post-reproductive mortality relationship and clarify our literature review.

o In Paragraph 2, we added the phrase “parity—post-reproductive mortality” (Line 65) to reference its association with childlessness on Line 66.18 Lines 67-69 now formally define infecundity and fertility (observed parity) based on definitions provided by (in an added citation) McFalls (2009). Descriptions of the childlessness-mortality association continue in Lines 70-71 and 80-83. 

o Paragraph 3, Lines 86-91, address childlessness and post-reproductive mortality, although the focus shifts to estimation of infecundity. 

o In the next section of the paper starting at Line 102 and Lines 104-108, we more fully describe the importance of childlessness in shaping the parity—post-reproductive mortality functional form. Specifically, we emphasize Hurt et al.’s (2007) finding that “in the thirteen historical cohorts studied, mortality was highest among childless and low-parity women and declined with parity; the contemporary birth cohorts more often exhibited U- or J shaped parity-mortality curves.”

o The remainder of the Childbearing Context section (to Line 165) develops why deeper investigation of childlessness, at least in the U.S. case, is important.

2. I also feel that the paper would benefit from closer attention to the historical context that the study cohorts were going through. In particular, I would have liked some recognition that the cohorts of women studied (born 1920-1941) were the cohorts going through childbearing during the baby boom. We know that such women had unusual marriage and childbearing patterns compared to those before and after. How might this affect conclusions?

Response: We appreciate these comments as they astutely highlight the importance of time (“period”) as context, and the importance of changing marriage and childbearing patterns across time. 

Change we made: In response to these comments, we have added the following text on Lines 115-123:

…Women in these historical birth cohorts shared atypical marriage and childbearing patterns; many could have given birth between 1946-1964, thereby contributing to the historical Baby Boom. Unlike women in contemporary birth cohorts, they more likely married, married prior to age 25, preferred to bear at least one child, had higher completed fertility, and primarily gave birth within marriage.7,29 Indeed, one advantage of studying historical populations marked by a high prevalence of early marriage and marital childbearing, is that the parity—post-reproductive mortality relationship is less likely to be masked. Another advantage is that the potential selectivity surrounding marital childbearing that occurs in contemporary birth cohorts is minimized.7,12 

This addition also answers the question: How might this affect conclusions? Our sample, comprised of women with a high prevalence of marriage, early marriage timing, and marital childbearing should facilitate assessment of the parity-mortality relationship, at least from a bio-evolutionary perspective.

3. Were there differences in the experience of the baby boom that affected black and white women differently? It might be easier to conceptualize these baby boom generations as an atypical episode rather than in between two ‘cross-overs’ (lines 123-126).

Response: We agree that marriage and childbearing patterns during the Baby Boom were historically atypical, with marked differences in patterns among Black and White women. Black women, until the mid-twentieth century, married at earlier ages, had higher marriage/remarriage rates than White women (were more exposed to marital fertility), and, at least 1900-1910, less likely practiced marital fertility control. Reynolds Farley and others found the high childlessness rates in 1880-1940 U.S. birth cohorts of Black women, born primarily in the Jim Crow South, worthy of note because their marital and childbearing patterns supported high fertility. As we noted above, in response to Reviewer 2’s second comment, we now acknowledge in the text just prior to the sentences referenced in this comment that all women in our study passed through some period of their childbearing career during that Baby Boom era, and that marriage and childbearing patterns during this historical period were atypical.

Changes we made: To address Reviewer 2’s additional obs

---

## [Decision Letter · Decision Letter 1]

19 Aug 2024

PONE-D-23-43696R1Parity and Post-Reproductive Mortality among U.S. Black and White Women: Evidence from the Health and Retirement StudyPLOS ONE

Dear Dr. Elman,

Thank you for submitting your manuscript to PLOS ONE. After careful consideration, we feel that it has merit but does not fully meet PLOS ONE’s publication criteria as it currently stands. Therefore, we invite you to submit a revised version of the manuscript that addresses the points raised during the review process.

Please respond to the reviewer's additional question.==============================

We look forward to receiving your revised manuscript.

Kind regards,

Emily W. Harville

Academic Editor

PLOS ONE

Journal Requirements:

Reviewers' comments:

Reviewer's Responses to Questions

**Comments to the Author**

1. If the authors have adequately addressed your comments raised in a previous round of review and you feel that this manuscript is now acceptable for publication, you may indicate that here to bypass the “Comments to the Author” section, enter your conflict of interest statement in the “Confidential to Editor” section, and submit your "Accept" recommendation.

Reviewer #1: All comments have been addressed

Reviewer #2: (No Response)

2. Is the manuscript technically sound, and do the data support the conclusions?

Reviewer #1: Yes

Reviewer #2: (No Response)

3. Has the statistical analysis been performed appropriately and rigorously? 

Reviewer #1: Yes

Reviewer #2: (No Response)

4. Have the authors made all data underlying the findings in their manuscript fully available?

Reviewer #1: Yes

Reviewer #2: (No Response)

5. Is the manuscript presented in an intelligible fashion and written in standard English?

Reviewer #1: Yes

Reviewer #2: (No Response)

6. Review Comments to the Author

Reviewer #1: Thank you for your thorough and thoughtful responses to the questions and comments the reviewers raised during the review process. I appreciate the care and attention you have given to addressing each point.

I have reviewed your revisions and responses and am satisfied with the changes made. I have no further comments or questions at this time.

Reviewer #2: Thank you for your careful responses to my previous questions. I have just one follow up question regarding my question 14:

My question and your response were:

14. Data. Regarding the observed parity and timing indicators (lines 301-311), age at first birth is

calculated using the age of participant and the age of their first child. This implies that

respondents were asked for the ages of all their children, including the age that any who had

died would be had they lived. This might be subject to more error than the ages of living

children, and it would be good to have an assessment of the effect of this on results.

Response: Although the HRS did query respondents about their number of children, it did not ask them

about the age(s) of all children born or list dates of all children born. Rather, it only asked them to report

the age of their oldest and youngest living child with whom they were in contact in the baseline

interview. The RAND family database harmonized this information. The codebook notes:

Age of the respondent’s youngest kid and oldest kid…variables are derived from the best guess

kid’s age in the Respondent-kid file. We noticed that some of the ages are over 80 years old.

Page 247, RAND HRS Family Data 2018 (V2). Santa Monica, CA (July 2023).

https://hrsdata.isr.umich.edu/sites/default/files/documentation/codebooks/1690235656/randhrsfam1992_2

018v2.pdf

Change we made: To clarify this we have added, Lines 337-339:

(The RAND family database provides information only about each participant’s oldest and

youngest currently-living child with whom they were in contact at the intake interview, not all

children born).

Follow-up question:

I appreciate you including this information, but I still think it is worth including a brief assessment of the possible effect of this on results. Clearly if their oldest or youngest child had died (or they had lost contact with them), then ages at first or last birth will be over or under-stated. Given that it is possible that the risk of child death is correlated with maternal health, how might this affect results?

7. PLOS authors have the option to publish the peer review history of their article (what does this mean?). If published, this will include your full peer review and any attached files.

Reviewer #1: **Yes: **Dovile Vilda

Reviewer #2: No

---

## [Author Response · Author response to Decision Letter 1]

3 Sep 2024

To the Editor and Reviewer: 

We are grateful to have the opportunity to respond to the one remaining Reviewer concern about our study. We describe this issue below and how we addressed the concern raised.

The original concern of the Reviewer was:

“14. Data. Regarding the observed parity and timing indicators (lines 301-311), age at first birth is calculated using the age of participant and the age of their first child. This implies that respondents were asked for the ages of all their children, including the age that any who had died would be had they lived. This might be subject to more error than the ages of living children, and it would be good to have an assessment of the effect of this on results.” 

 Our original response was: 

 Although the HRS did query respondents about their number of children, it did not ask for the age(s) of all children born or list dates of all children born. Rather, it only asked them to report the age of their oldest and youngest living child with whom they were in contact in the baseline interview....

 We appreciate the chance to clarify our initial response to the Reviewer’s original concern. We apologize; we should have made it clearer that the age of birth variables that we used were indeed based on living children in-contact with the respondent, as obtained from the RAND Family Respondent File. The HRS did not ask respondents to estimate the age that any who had died would be had they lived.

Please let us further note that there is a measurement gap in HRS/RAND data, regarding: (1) parity/number of children ever born; and (2) number of own in-contact living children as ascertained in late-life (at baseline). Regarding the former, HRS/AHEAD directly queried respondents at intake about their number of children born (RAND variable, RAEVBRN). Regarding the latter, the RAND Family File compiled indirect information about respondents’ number of own currently-living children from household rosters (Family File variable, R_OWNKIDKN) that captured children living in households, and respondent responses to queries about children living outside of households. HRS/AHEAD respondents were never asked to report their number of children surviving, in the context of their number of children ever born (RAEVBRN). 

We reiterate, in response to the Reviewer’s initial concern, that we calculated age at birth variables based on the living (surviving) oldest child age reported in the RAND Family File. 

In response to our initial response, the Reviewer raised a further concern: 

“I appreciate you including this information, but I still think it is worth including a brief assessment of the possible effect of this on results. Clearly if their oldest or youngest child had died (or they had lost contact with them), then ages at first or last birth will be over or under-stated. Given that it is possible that the risk of child death is correlated with maternal health, how might this affect results?”

We address this new concern here.

The Reviewer is concerned that age at birth will be over or under-stated if an oldest/youngest child had died (or lost contact). This is an interesting and important possibility to consider. However, as noted above, HRS/RAND data do not provide respondents’ total number of child deaths or the deaths/death dates of any children, in or out of the household. As the Reviewer notes, among alive in-contact children in the RAND Family File, one does not know the true birth order of the reported child (i.e., whether the oldest/youngest child was truly so). One also does not know whether a gap between women’s number of children born (RAEVBRN) and their number of later-life in-contact own living children (R_OWNKIDKN) captures birth child(ren) who: died in infancy or in adulthood; were given up at birth for adoption (not uncommon in these birth cohorts); lost contact due to moving overseas (pre-internet, satellite world) or due to parent-child conflict.

With this recognition, we developed perhaps the only test possible of whether the use of RAND Family Respondent File age of oldest living child might introduce error in findings, contingent on whether even earlier-born child(ren) had died/lived/kept contact. We created a Flag variable, using both the RAEVBRN number of children ever born and RAND Family File variable (R_OWNKIDKN) number of own alive, in-contact children in late-life (at baseline), coded as follows:

Flag=1 A respondent is coded 1 if her reported number of children born differs from her number of own in-contact living children as compiled in the RAND Family File. 

Flag=0 A respondent is coded 0 if her reported number of children born is equal to (matches) the number of own in-contact living children as compiled in the RAND Family File. 

Again, HRS/RAND data do not allow us to examine birth/survival data for each child born; deaths/death dates are not reported. But we believe that the Flag at least partly addresses the question of whether age of birth findings might be contingent on whether a possible “true” first/oldest child died/lost contact. Such cases would be included in the Flag=1 subgroup. We caution however that, among women whose number of children born is not equal to their number of in-contact alive children in late-life (Flag=1), missing children) could still be of any birth order (not just oldest or youngest).

Our sensitivity analysis adds this Flag to Models for parous women (Models 5-7) in Tables 3-5. The Flags are not necessary in the nulliparous/parous (or all-women) parity models (Tables 3-5, Models 1-4) because they do not include the age at birth variables: nulliparous women have missing values for all age at birth measures. Our statistical findings in the attached table show that virtually all age at first birth hazard ratios (Models 5-7, Tables 3-5) do not change in strength or significance. Therefore, the answer is “No” to the question of whether coding of Age of Birth, using the live, in-contact children matters for findings; findings do not differ when we account for kids who had died/lost contact (i.e., the gap between RAEVBRN and R_OWNKIDKN). But better (different) data are required to ascertain directly whether it is possible that the risk of child death is correlated with maternal health or longevity. HRS respondents’ children ever born (RAEVBRN) could be “gone” by later life due to social, interpersonal, or health/survival processes. The data do not allow testing of these pathways. 

Changes we made: we added the attached sensitivity analysis to the Appendix, as Supplemental Table S6, and added the following to the Analytical Strategy section, starting on Line 503: 

Information about children ever born, harmonized in RAND, was directly reported by HRS/AHEAD participants. In contrast, the RAND Family Respondent File ascertained participants’ number of in-contact, alive children from household and participant records. As HRS/RAND data do not report children surviving in the context of children ever born, and RAND Family data do not report deaths/death dates of children who died prior to HRS/AHEAD at baseline, there is a measurement gap: given the universe of children ever born, we only have information about surviving children in late-life (RAND Family Respondent File). This gap suggests that, among parous women, age of oldest child in RAND Family data may be mis-stated. This could introduce error in age at birth findings, contingent on whether even earlier-born child(ren) had died/lost contact (i.e., were not present in the RAND Family roster). In a sensitivity analysis (see Supplement Table S6), we examined possible measurement error in age at first birth variables used in Models for parous women by creating a Flag variable that differentiated number of children ever born and number of own alive, in-contact children in late-life. The Flag=1 applied if women’s number of children born differed from their number of in-contact, living children in late-life or =0 if the number of children matched. We found that statistical results (the size and significance of hazard ratios) for age at first birth variables did not change when Flags were included in Models. 

Again, we are grateful for the careful attention paid to our paper and for the chance to revise it. We believe we have been able to respond to the concerns raised and that our paper is stronger as a result.

---

## [Editor Report · Decision Letter 2]

5 Sep 2024

Parity and Post-Reproductive Mortality among U.S. Black and White Women: Evidence from the Health and Retirement Study

PONE-D-23-43696R2

Dear Dr. Elman,

We’re pleased to inform you that your manuscript has been judged scientifically suitable for publication and will be formally accepted for publication once it meets all outstanding technical requirements.

Kind regards,

Emily W. Harville

Academic Editor

PLOS ONE
---

## [Editor Report · Acceptance letter]

10 Sep 2024

PONE-D-23-43696R2 

PLOS ONE

Dear Dr. Elman, 

I'm pleased to inform you that your manuscript has been deemed suitable for publication in PLOS ONE. Congratulations! Your manuscript is now being handed over to our production team.

Kind regards, 

on behalf of

Dr. Emily W. Harville 

Academic Editor

PLOS ONE